**Title:** High metabolism and periodic hypoxia associated with drifting macrophyte detritus in the
shallow subtidal Baltic Sea
**Author list:** Karl M. Attard[1,2,3], Anna Lyssenko[3], Iván F. Rodil[3,4]
**Corresponding author:** Karl M. Attard karl.attard@biology.sdu.dk
**Author affiliations:**
[1] Department of Biology, University of Southern Denmark, 5230 Odense M, Denmark
[2] Danish Institute for Advanced Study, University of Southern Denmark, 5230 Odense M, Denmark
[3] Tvärminne Zoological Station, University of Helsinki, J.A. Palménin tie 260, 10900 Hanko,
Finland
[4] Department of Biology (INMAR), Faculty of Marine and Environmental Sciences, University of
Cádiz, Puerto Real, Spain
**Keywords:** benthic ecosystems, primary production, respiration, oxygen fluxes, biodiversity
**Abstract**
Macrophytes form highly productive habitats that export a substantial proportion of their primary
production as particulate organic matter. As the detritus drifts with currents and accumulates in
seafloor depressions, it constitutes organic enrichment and can deteriorate $O_2$ conditions on the
seafloor. In this study, we investigate the $O_2$ dynamics and macrobenthic biodiversity associated
with a shallow ~2300 $m^2$ macrophyte detritus field in the northern Baltic Sea. The detritus,
primarily *Fucus vesiculosus* fragments, had a biomass of ~1700 g dry weight $m^{-2}$, approximately
1.5-fold larger than nearby intact *F. vesiculosus* canopies. A vertical array of $O_2$ sensors placed
within the detritus documented that hypoxia ($[O_2] < 63$ µmol $L^{-1}$) occurred for 23% of the time and
terminated at the onset of wave-driven hydrodynamic mixing. Measurements in five other habitats
nearby spanning bare sediments, seagrass, and macroalgae indicate that hypoxic conditions were
unique to detritus canopies. Fast-response $O_2$ sensors placed above the detritus documented pulses
of hypoxic waters originating from within the canopy. These pulses triggered a rapid short-term (~5
min) deterioration of $O_2$ conditions within the water column. Eddy covariance measurements of $O_2$
fluxes indicated high metabolic rates with daily photosynthetic production offsetting up to 81 % of
the respiratory demands of the detritus canopy, prolonging its persistence within the coastal zone.
The detritus site had a low abundance of crustaceans, bivalves, and polychaetes when compared to
other habitats nearby, likely because their low-$O_2$ tolerance thresholds were often exceeded.
**1. Introduction**
Oxygen availability determines ecosystem health and the biogeochemical function of coastal waters
(Diaz and Rosenberg, 2008; Middelburg and Levin, 2009; Breitburg et al., 2018). When in gaseous
equilibrium with air, seawater typically contains an $O_2$ concentration ($[O_2]$) between 200-400 μmol
$L^{-1}$, depending on the water temperature and the salinity (Garcia and Gordon, 1992). However, both
abiotic and biotic processes cause significant departures from equilibrium. The main source of $O_2$ to
coastal waters is the atmosphere, where the diffusion of $O_2$ is governed by the air-to-sea gas
exchange rate (Berg and Pace, 2017; Long and Nicholson, 2018). In shallow waters and light-
exposed seafloor sediments, $O_2$ is produced by primary producers as a by-product of
photosynthesis, and it is consumed by consortia of microbes and fauna directly, through aerobic
respiration, and indirectly, through the oxidation of reduced substances (Glud, 2008). If $O_2$
consumption exceeds supply for a sufficiently long period, $O_2$ conditions deteriorate and become
hypoxic ($[O_2] < 63$ μmol $L^{-1}$). Hypoxia is becoming more common, more intense, and is affecting
larger areas of coastal waters, increasingly placing ecosystems and the services they provide at risk
(Breitburg et al., 2018). There are several well-known variants of coastal hypoxia (Diaz and
Rosenberg, 2008; Carstensen and Conley, 2019). Seasonal hypoxia, the most common form,
typically occurs in summer when warm waters, strong stratification, and high organic enrichment
combine to deplete $O_2$ until autumn (Robertson et al., 2016). Periodic hypoxia, in contrast, occurs
more often due to local weather dynamics and tidal cycles but individual events are shorter (Diaz
and Rosenberg, 1995), whereas diel cycles with large day-to-night $[O_2]$ excursions trigger hypoxia
for a few hours daily (Davanzo and Kremer, 1994; Tyler et al., 2009). All events are expected to
affect biodiversity and biogeochemical cycling to varying degrees. Seasonal and periodic hypoxia
are associated with large-scale mortality of organisms and a switch between retention and removal
of bioavailable nutrients such as nitrate, ammonium, phosphate, and toxic hydrogen sulfide
(Middelburg and Levin, 2009; Carstensen and Conley, 2019). Short-term hypoxia can similarly
exceed lethal and non-lethal thresholds for many taxa (Vaquer-Sunyer and Duarte, 2008), although,
due to their sporadic nature, their occurrence and impacts are less understood.
Given the importance of $O_2$ in coastal waters, $[O_2]$ is one of the most frequently measured
environmental parameters. Near-seabed $[O_2]$ is typically measured using long-term stable $O_2$

sensors (e.g. optodes) (Bittig et al., 2018) that are moored ~0.3-1.0 m above the seafloor, or by performing vertical profiles of water column [$O_2$] down to ~1.0 m above the seafloor using multiparameter sondes. National monitoring programs such as those maintained by the Swedish Meteorological and Hydrological Institute and the Finnish Environment Institute provide a wealth of essential open-access data, enabling important analyses detailing the prevalence and intensity of coastal hypoxia (Virtanen et al., 2019; Conley et al., 2011; Carstensen and Conley, 2019). Notwithstanding the progress being made in coastal monitoring, it was demonstrated more than 40 years ago that the largest [$O_2$] gradients may occur just a few cm above the seafloor due to the high reactivity of marine sediments and a strong benthic $O_2$ demand (Jorgensen, 1980). To date, records of hypoxia in the shallow subtidal zone are still somewhat scarce. In a compilation of monitoring data for the northern Baltic Sea (Gulf of Finland and Archipelago Sea), Virtanen et al. (2019) found that just 11 out of 461 (or 2.4%) of the monitoring stations that registered hypoxia occurred in waters < 5 m depth. While this may reflect a true signal that hypoxia is more widespread in deeper coastal waters, it is also likely that hypoxic conditions go undetected if measurements are performed away from the seafloor, as is common practice (Conley et al., 2011; Virtanen et al., 2019).

Around two-thirds of the ocean's photosynthetic biomass is bound in macrophytes growing in shallow waters along the world's coastline (Smith, 1981). Through seasonal decay, epiphyte growth, grazing, and physical forcing (e.g. waves, currents, ice scouring), macrophytes export a large proportion of their primary production (~40 %) to their surroundings as detritus (Attard et al., 2019a; Krumhansl and Scheibling, 2012; Duarte and Cebrián, 1996). Macrophyte detritus drifts with the currents and accumulates on the shoreline and in low-energy marine environments (e.g. shallow seafloor depressions and in deeper waters), where it constitutes habitat structure and organic enrichment to the receiving habitat (Norkko and Bonsdorff, 1996b). Given high enough abundance, detritus suppresses the diffusion of $O_2$ from the water column to the sediment surface and it exacerbates $O_2$ depletion on the seabed as it decays. Large accumulations of unattached ephemeral macroalgae such as the brown algae *Ectocarpus siliculosus* and *Pylaiella littoralis* are common in eutrophic coastal waters such as the Baltic Sea, forming thin mats above the seafloor typically a few centimeters thick (Norkko and Bonsdorff, 1996a). While coastal hypoxia is most commonly associated with eutrophic waters such as the Baltic Sea (Carstensen and Conley, 2019; Conley et al., 2011), hypoxic (and even sulfidic) conditions have been reported in remote and more pristine environments such as the high Arctic due to large accumulations of detritus produced from

perennial brown seaweeds (Glud et al., 2004). However, the $O_2$ dynamics within accumulations of drifting detritus and the potential implications for the associated fauna remain poorly understood. Understanding the ecological and biogeochemical implications of drifting macrophyte detritus is particularly important given the ambitions to vastly increase macroalgal farming (Broch et al., 2019), which would result in increased deposition of macrophyte detritus on the coastal seafloor (Broch et al., 2022).

In this study, we investigate the $O_2$ dynamics and macrobenthic biodiversity associated with a shallow ~2300 $m^2$ macrophyte detritus field composed of *Fucus vesiculosus* fragments in the northern Baltic Sea. To assess $O_2$ production versus consumption rates of the detritus canopy, we deployed an eddy covariance system on multiple occasions to extract benthic $O_2$ fluxes non-invasively. Using a vertical array of $O_2$ sensors and an acoustic velocimeter, we monitored $O_2$ distribution within the canopy and the hydrodynamics above the canopy to assess the occurrence and intensity of hypoxic events and their links to local hydrodynamics. We performed biodiversity surveys to identify the prevailing taxa, and we compared hypoxic thresholds of these taxa to $[O_2]$ measured *in situ* to identify potential stress. Measurements were also performed in five other habitats nearby spanning bare sediments, seagrass, and macroalgae for comparison.

**2. Materials and Methods**

*2.1. Study location*

The study was performed in the microtidal Baltic Sea nearby the Tvärminne Zoological Station in SW Finland. Although the focus of our study was to investigate drifting macrophyte detritus, we selected an additional five study sites within the shallow subtidal zone (2-4 m depth) for comparison, representing key habitats in the Baltic Sea: one site with bare sediments, two sites with seagrass (predominantly *Zostera marina*; sheltered and exposed), and two sites with intact macroalgae canopies (predominantly *Fucus vesiculosus*; sheltered and exposed) (Table 1).

| Site | Location | Deployment start | Deployment duration (h) | Water depth (m) | Water temperature (°C) | Minimum $O_2$ (µmol $L^{-1}$) | Maximum $O_2$ (µmol $L^{-1}$) | Hypoxia duration (h) |
|---|---|---|---|---|---|---|---|---|
| Macrophyte detritus | 59 811613 N 23 206624 E | 29-05-2018 | 120 | 3.0 | 12 | 0.6 | 429 | 27 |
| Bare sediments | 59 841532 N 23 253370 E | 20-05-2018 | 96 | 3.7 | 11 | 307 | 407 | 0 |
| Sheltered *Z. marina* | 59 841551 N 23 251203 E | 27-05-2018 | 87 | 4.0 | 16 | 272 | 333 | 0 |

| Exposed *Z. marina* | 59 827008 N 23 151976 E | 08-06-2018 | 120 | 2.9 | 10 | 281 | 437 | 0 |
|---|---|---|---|---|---|---|---|---|
| Sheltered *F. vesiculosus* | 59 826856 N 23 209721 E | 08-06-2018 | 120 | 2.0 | 10 | 253 | 489 | 0 |
| Exposed *F. vesiculosus* | 59 811359 N 23 207281 E | 31-05-2018 | 116 | 2.0 | 9 | 287 | 427 | 0 |

Table 1: Environmental conditions and low-oxygen events at the six study sites

*2.2. [O₂] dynamics in benthic habitats*

To investigate the near-bed [$O_2$] dynamics and its environmental controls, we equipped a tripod frame with a suite of sensors consisting of three cross-calibrated dissolved [$O_2$] loggers with inbuilt temperature compensation (HOBO U26-001, Onset), a 6 MHz acoustic velocimeter (Vector, Nortek), a photosynthetic active radiation (PAR) sensor (RBRsolo with Licor PAR Quantum 192SA), and a saltwater conductivity sensor (HOBO U24-002-C). The [$O_2$] loggers have a factory-specified accuracy of $\pm 6$ µmol L$^{-1}$ from 0 to 250 µmol L$^{-1}$, $\pm 16$ µmol L$^{-1}$ from 250-625 µmol L$^{-1}$, a resolution of 0.6 µmol L$^{-1}$ and a 90% response time ($T_{90}$) < 2 min. The [$O_2$] and conductivity sensors were mounted onto a 75 cm-long stainless-steel rail affixed to the tripod leg (Fig. 1). The sensors were secured to the rail at various heights above the seabed using rail mount clamps. For the study sites with canopies, two sensors were set inside the canopy; one sensor was ~5 cm above the seafloor and one was close to the top of the canopy (15-25 cm). The third sensor was placed in the water above the canopy (~35 cm above the seafloor). The tripod was deployed by divers from a small boat and was carefully positioned on the seafloor using a lift bag. The exact sensor heights were noted by the divers once the instrument was on the seafloor. The instrument was left to record data for 3-5 days at each site. The velocimeter sampled three-dimensional flow velocity continuously at 8 Hz, whereas the [$O_2$], temperature, conductivity, and PAR sensors recorded data every minute.

To investigate [$O_2$] dynamics and its environmental drivers, all sensor time series were aligned in time and analyses were performed to investigate vertical gradients in O₂ distribution, diel [$O_2$] excursions, and boundary-layer hydrodynamics. We assessed the occurrence of hypoxia ([$O_2$] < 63 µmol L$^{-1}$) by quantifying the magnitude (lowest [$O_2$] value) and the duration (in hours) of hypoxic events. The high-frequency velocity data were used to calculate mean flow velocity magnitude ($\overline{U}$) as the sum of streamwise ($u$) and traverse ($v$) components, as $\overline{U} = \sqrt{u^2 + v^2}$.

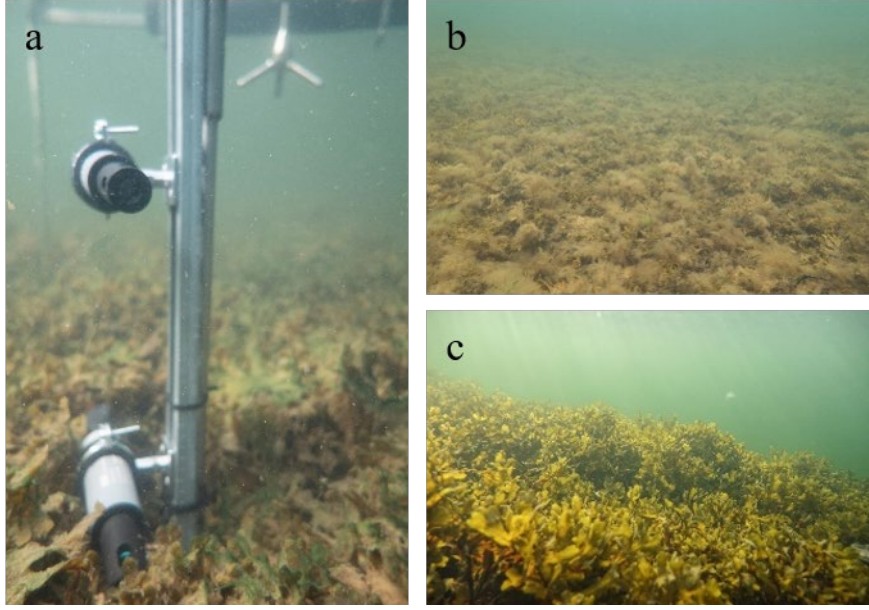


Fig. 1: The study area showing (a) the instrument deployed within the detritus canopy, (b) a broad-
scale view of the detritus accumulation area, and (c) a nearby intact *Fucus vesiculosus* canopy.
*2.3. Benthic $O_2$ fluxes*
An aquatic eddy covariance system was deployed at the detritus site to quantify benthic $O_2$ fluxes at
the canopy-water interface on three occasions (June 2017, September 2017, and May 2018). Eddy
covariance integrates over a relatively large seafloor area (typically ~30 m$^2$) (Berg et al., 2007), and
extracts fluxes without disturbing the hydrodynamics or the light, which is particularly useful when
trying to understand the mechanistic drivers of [$O_2$] dynamics (Berg et al., 2022). The eddy
covariance setup was identical to the tripod frame described above, with the addition of a fast-
response ($T_{90} < 0.3$ s) [$O_2$] microsensor setup for covariance measurements (McGinnis et al., 2011).
The hardware and data processing techniques are described in detail in Attard et al. (2019b). This
instrument can capture the entire range of flux-contributing turbulent eddies within the benthic
boundary layer, and this information is used to approximate the benthic $O_2$ flux non-invasively
(Berg et al., 2003; Berg et al., 2022). The instrument recorded co-located measurements of the
vertical velocity ($w$) and the $O_2$ concentration ($C$) at 32 Hz, and the data were processed using a
multiple-step protocol detailed in Attard et al. (2019b) to extract and quality-check benthic fluxes.
In short, the data streams for $w$ and $C$ were decomposed into mean and fluctuating components
using Reynolds decomposition, as $w = \bar{w} + w'$ and $C = \bar{C} + C'$ (Berg et al., 2003). The turbulent
flux ($J_{EC}$) was then computed in units of mmol $O_2$ m$^{-2}$ h$^{-1}$ as $J_{EC} = \overline{w'C'}$, where the overbar
represents a period of 15 min. The turbulent flux was then summed with a storage correction term to
calculate the total benthic flux ($J_{benthic}$, mmol $O_2$ m$^{-2}$ h$^{-1}$) (Rheuban et al., 2014), as:

$$J_{benthic} = J_{EC} + \int_0^h \frac{\partial C}{\partial t} \, \mathrm{d}z$$

The storage correction term was defined using the three [$O_2$] optodes placed within and above the
canopy. For the correction, we defined a matrix with the number of rows $n$ corresponding to the
sensor measurement height above the seafloor (1 row per cm) (Camillini et al., 2021). To do this,
the oxygen time series, consisting of [$O_2$] measurements performed at three heights within the
canopy, were converted to a matrix using the software package OriginPro 2022. Since the
measurement height of the three sensors were spaced nonlinearly, the data were first converted to
XYZ column format using the w2xyz function. Next, the three rows, representing the [$O_2$] time
series measurements at three heights, were expanded to $n$ rows, with $n$ representing the sensor
measurement height in cm (from 0 to $n$ cm above seabed, 1 row per cm) using the XYZ Gridding
function. This generated a matrix of $n$ rows consisting of linearly interpolated [$O_2$]. Interpolation
was performed using the Random (Renka Cline) gridding method. Next, a storage correction term
was calculated for each 1 cm cell as described by Rheuban et al. (2014), and the total storage
correction was subsequently computed for the water volume below the sensor measurement height
as the sum of the $n$ rows. The high-frequency [$O_2$] time series from the fast-response microsensors
were also analyzed to identify any pulses of low [$O_2$] waters originating from within the canopy and
propagating upwards into the water column.
*2.4. Benthic metabolic rates*
The $O_2$ flux time series was separated into individual 24 h periods (midnight to midnight). The
daytime flux (Flux$_{day}$, mmol $O_2$ m$^{-2}$ h$^{-1}$) was computed as a bulk average of fluxes measured when
PAR > 1.0 µmol m$^{-2}$ s$^{-1}$. The nighttime flux (Flux$_{night}$, mmol $O_2$ m$^{-2}$ h$^{-1}$) was calculated as the
average of the remaining fluxes, when PAR < 1.0 µmol m$^{-2}$ s$^{-1}$. These two values and the number of
daylight hours ($h_{day}$) were used to estimate the daily photosynthetic rate, termed the gross primary
production (*GPP*, in mmol $O_2$ m$^{-2}$ d$^{-1}$), as $GPP = Flux_{day} + abs(Flux_{night}) * h_{day}$, and daily
respiration (*R*, in mmol $O_2$ m$^{-2}$ d$^{-1}$), as $R = abs(Flux_{night}) * 24$, assuming a light-independent
respiration rate. The latter is a common assumption, but it is known that it underestimates the true
metabolic activity (Fenchel and Glud, 2000; Juska and Berg, 2022). The daily balance between
*GPP* and *R*, termed the net ecosystem metabolism (*NEM*, in mmol $O_2$ m$^{-2}$ d$^{-1}$) was estimated as
$NEM = GPP - R$ (Attard et al., 2019b).
The relationship between seafloor PAR and the in situ benthic $O_2$ flux was investigated using light-
saturation curves. Hourly $O_2$ fluxes were plotted against the corresponding near-bed incident PAR
and the relationship between the two was investigated using a modified tangential hyperbolic
function by Platt et al. (1980), as $O_2\ flux = P_m * \tanh\left(\frac{\alpha I}{P_m}\right) - R$, where $P_m$ is the maximum rate of
hourly gross primary production, $\alpha$ is the initial quasi-linear increase in $O_2$ flux with PAR, $I$ is near-
bed irradiance (PAR), and $R$ is the dark respiration rate. The photosaturation parameter, $I_k$ (μmol
PAR m$^{-2}$ s$^{-1}$) was derived as $P_m/\alpha$. Non-linear curve fitting was performed in OriginPro 2022 using
a Levenberg–Marquardt iteration algorithm, until a Chi-Squared tolerance value of 1E-9 was
reached (Attard and Glud, 2020).
*2.5. Biodiversity sampling*
At all six sites, we aimed to obtain a quantitative understanding of the abundance, biomass, and
species richness of macrophytes and macrofauna (infauna and epifauna). The different habitats
required different sampling strategies, since four sites were sedimentary (bare sediments site, two
seagrass sites, and the detritus sites) and two sites were rocky (two macroalgal sites) (Rodil et al.,

207   2019).

At the time of our study, the detritus site had a ~20-cm thick detritus mat covering the seabed
sediments. The detritus canopy was sampled using large stainless steel core liners (inner diameter =
19 cm; $n = 4$) capable of cutting through the mat, and the collected samples were transferred into a
fine-mesh bag. In the laboratory, the detritus was rinsed through a 0.5 mm sieve to collect the
associated epifauna. Samples of algal detritus were dried at 60°C for 48 hours and the biomass was
calculated as dry weight /m$^2$.
Macroinfauna at the four sedimentary habitats was sampled using six sediment cores (inner
diameter = 5.0 cm, depth = 15 cm). The samples were sieved through a 0.5 mm sieve and animals
were stored in alcohol for later identification. At the seagrass sites, representative macrophyte
samples were collected by divers from an area around the tripod frame at the end of the deployment
using four randomly-placed quadrats (20 x 20 cm). The seagrass within each quadrat was gently
uprooted and was transferred into a net-bag. In the laboratory, the samples were rinsed through a
0.5 mm sieve to collect all the associated epifauna. The animals were stored in alcohol for later

identification, and the seagrass was frozen in sealed bags for further processing. The seagrass samples were later thawed, and individual shoots were counted to determine the canopy density in $m^2$. The above- and below-ground macrophyte biomass was separated, dried at 60°C for 48 hours and weighed.

At the rocky sites, *F. vesiculosus* individuals ($n = 4$) were randomly collected from around the instrument in fine-mesh bags. Randomly-placed quadrats (1 $m^2$, $n = 4$) were used to quantify the number of *F. vesiculosus* individuals per $m^2$. At the laboratory, the collected *F. vesiculosus* samples were carefully rinsed through a 0.5 mm sieve to collect the epifauna. The height of the *F. vesiculosus* canopy was determined from the average length of the sampled individuals. Both *F. vesiculosus* and epiphytes were separated to the extent possible, dried at 60 °C for 48 h and weighed. To collect any macrofauna on the bare rock beneath the *F. vesiculosus* canopy, Kautsky-type samplers were placed on the seafloor and the 20 cm x 20 cm area was gently scraped using a spoon into a fine-mesh sampling bag. In the laboratory, all the macrofauna from the four replicates were sieved through a 0.5 mm sieve and stored in alcohol.

The fauna from all habitats was sorted, identified to species level, counted, and weighed. The wet weight for each species was noted with 0.0001 g accuracy. In cases where the fauna occurred in very high numbers, the sample was placed in a water-filled tray and divided into eight sectors. Four sectors were randomly chosen to calculate abundance and biomass. The length of gastropods and bivalves was measured from anterior to posterior axis using Vernier callipers (accuracy = 0.01 mm) for conversion to ash-free dry mass (AFDM). The AFDM of bivalves and gastropods was calculated using established relationships between length and weight for Baltic Sea fauna (Rumohr et al., 1987).

The abundance (ind $m^{-2}$) and biomass (AFDM/SFDM g $m^{-2}$) of the invertebrates across sites were calculated. Primer (v.7 and PERMANOVA+) software was used to perform the nonmetric multidimensional scaling (nMDS, with fourth-root-transformed data) to visualize macrofauna assemblages between sites. ANOSIM based on the Bray-Curtis similarity matrix was also performed in Primer (site as a fixed factor, 4999 random sample permutations) to compare differences in macrofauna abundance and biomass between sites.

## 3. Results

*3.1. Environmental conditions*

Average water depth ranged from 2.0 m to 4.0 m at the six study sites, and average water
temperature ranged from 9 °C to 16 °C during the study period (Table 1). Hypoxic conditions were
only detected at the detritus site. Bottom-water [$O_2$] at the detritus site ranged from 1 µmol L$^{-1}$ to
429 µmol L$^{-1}$, with hypoxic conditions occurring for 27 h out of the 120 h long deployment (i.e. for
23 % of the time) (Table 1). At the five other measurement sites, [$O_2$] were well above hypoxic
conditions, with overall concentrations following diel patterns and ranging from 250 µmol L$^{-1}$ to
490 µmol L$^{-1}$ (Table 1).
*3.2. [$O_2$] dynamics in detritus canopies*
The [$O_2$] measurements within the detrital canopy document a highly dynamic [$O_2$] environment
driven by light availability and flow velocity (Fig. 2). Within the upper layers of the canopy (i.e.
~10 to 25 cm above the seafloor), [$O_2$] and temporal dynamics largely follow diel patterns driven by
light availability, with large ~250 µmol L$^{-1}$ diel excursions in [$O_2$]. There, the [$O_2$] was lowest in
the morning (~160 µmol L$^{-1}$) and highest in the evening (~430 µmol L$^{-1}$). In all cases, [$O_2$] within
the upper canopy region was above hypoxic thresholds. However, under low average flow
velocities < 2 cm s$^{-1}$, [$O_2$] within the lower canopy region (< 10 cm) deviated substantially from the
conditions above. No diel variations in [$O_2$] were observed during these periods, and [$O_2$] rapidly
became hypoxic for sustained periods (> 24 h long), with [$O_2$] being very low (< 10 µmol L$^{-1}$)
during ~10 hr (~8 % of the time) (Fig. 2). As hypoxia persisted throughout the night under low flow
velocities, low [$O_2$] extended upwards into the canopy. Hypoxic conditions ended at the onset of
higher mean flow velocities of ~7 cm s$^{-1}$, which initiated a rapid (i.e. within 1.5 hr) oxygenation of
the entire canopy.

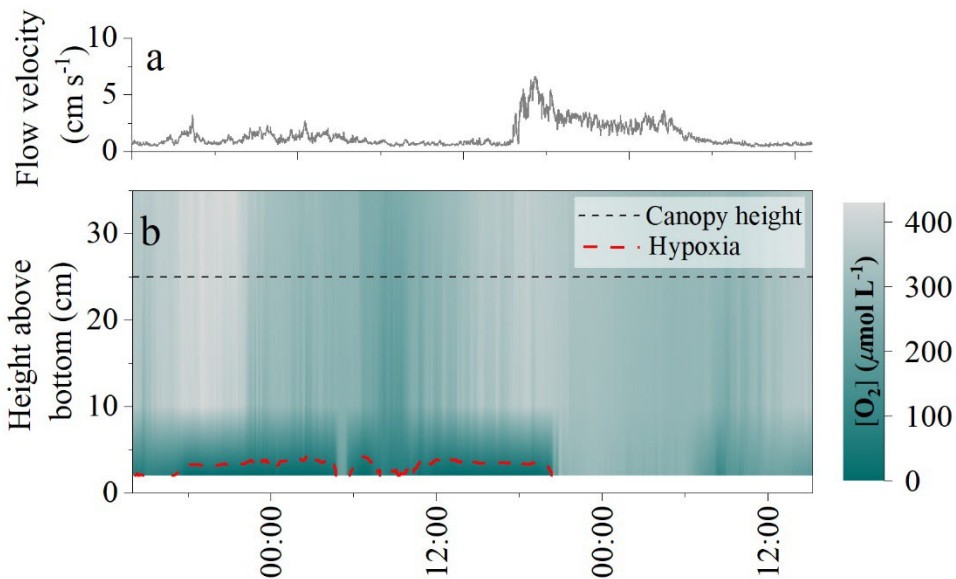


Fig. 2: (a) Flow velocity measured by the velocimeter 10 cm above the detritus canopy and (b) $O_2$ distribution within the canopy as resolved by three $O_2$ sensors located at 3 cm, 10 cm, and 35 cm above the seafloor. Deployment starting from 29th May 2018.

*3.3. Pulses of hypoxic waters*

High-frequency [$O_2$] measurements performed 10 cm above the detritus canopy document transient pulses of hypoxic water originating from within the canopy and propagating upwards into the water column (Fig 3). Such pulses typically followed quiescent weather and occurred at the onset of increased flow velocities. It took < 1 min to reduce [$O_2$] in the water column from 220 µmol L$^{-1}$ to 65 µmol L$^{-1}$. Subsequently, a recovery period followed where [$O_2$] gradually increased back to previous concentrations over a ~5 min period. These rapid variations in water column [$O_2$] were not captured by the slow-response [$O_2$] optode sampling at 1 min intervals.

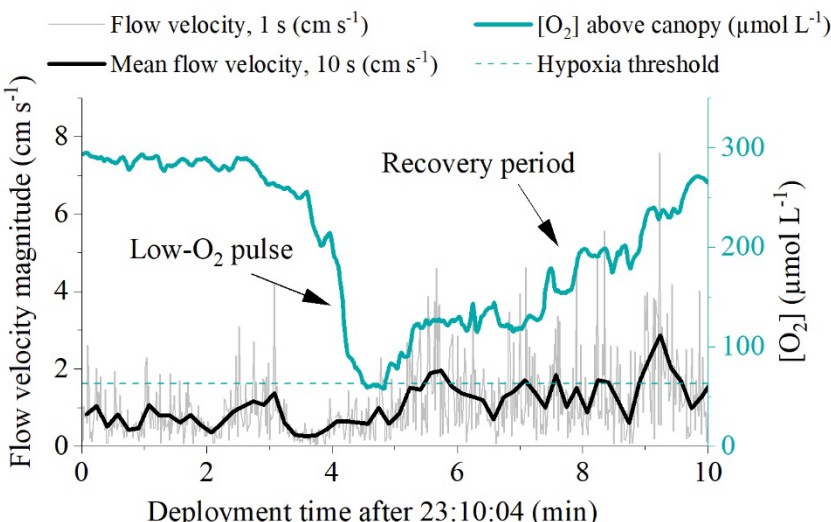

Figure 3: High-frequency [$O_2$] measured 10 cm above the detrital canopy documented pulses of hypoxic water originating from within the canopy and propagating upwards into the water column. Data from 20$^{th}$ September 2017.

### 3.4. Benthic $O_2$ fluxes and detritus metabolic rates

The eddy covariance measurements at the detritus site produced three days of continuous flux data in June 2017, three days of data in September 2017, and five days of data in May 2018. Benthic $O_2$ fluxes documented a dynamic $O_2$ exchange rate driven by light availability and flow velocity. During quiescent periods with low flow velocity < 2 cm s$^{-1}$, a clear diel signal in the $O_2$ flux was observed, indicating substantial primary production associated to the detritus canopy. Higher flow velocities stimulated $O_2$ uptake rates by up to 5-fold, indicating that canopy ventilation through mixing increased $O_2$ uptake.

Hourly $O_2$ fluxes ranged from -22 mmol $O_2$ m$^{-2}$ h$^{-1}$ at night to 13 mmol $O_2$ m$^{-2}$ h$^{-1}$ during the day and showed a distinct diel cycle in response to sunlight availability (Fig. 4). Daily R ranged from 26 to 97 mmol $O_2$ m$^{-2}$ d$^{-1}$, and daily GPP was between 15 and 74 mmol $O_2$ m$^{-2}$ d$^{-1}$. Daily R exceeded GPP in all 11 measurement days (net heterotrophic), with NEM ranging from -7 to -32 mmol $O_2$ m$^{-2}$ d$^{-1}$ (Fig. 4, Table A1). The deployment average ($\pm$ SD) GPP:R for the detritus canopy was 0.77 $\pm$ 0.04 in June 2017 ($n$ = 3), 0.55 $\pm$ 0.02 in September 2017 ($n$ = 3), and 0.77 $\pm$ 0.00 in May 2018 ($n$ = 5), and the global mean GPP:R was 0.71 $\pm$ 0.11 (n = 11).

There was a significant positive relationship between daily detritus GPP and R in all measurement campaigns, with the detritus canopy seemingly becoming more heterotrophic (i.e. R > GPP) as the

magnitude of the metabolic rates increased (Fig. 5, Table A1). Significant positive relationships were also observed between daily detritus GPP and daily seabed PAR (Table A1). Canopy light-use efficiency (LUE), estimated as the ratio between daily GPP and daily PAR (Attard and Glud, 2020), was 0.004 $O_2$ photon$^{-1}$ in June 2017, 0.006 $O_2$ photon$^{-1}$ in September 2017, and 0.004 $O_2$ photon$^{-1}$ in May 2018 (Table A1).

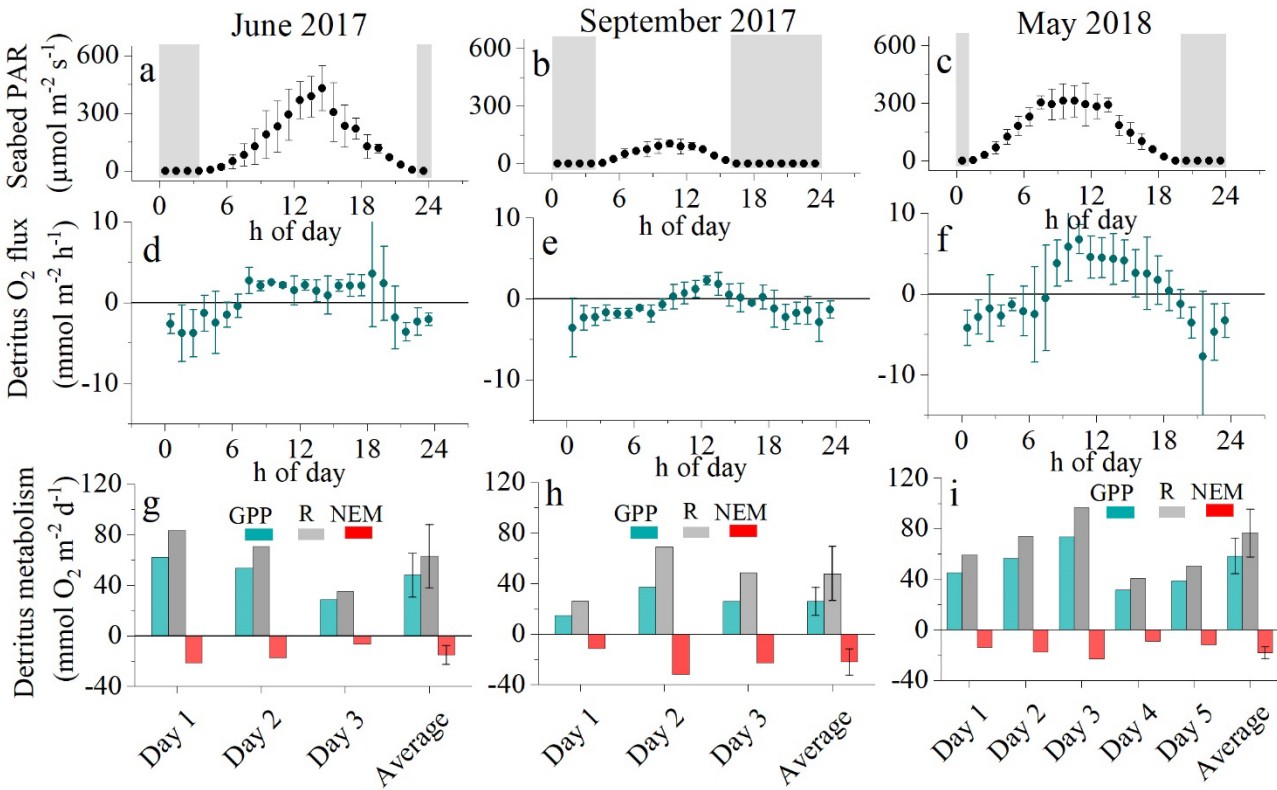

Fig. 4: Hourly seabed PAR (a, b, c) with night-time periods grey shaded, hourly $O_2$ fluxes (d, e, f) and daily metabolism estimates of gross primary production (GPP), respiration (R), and net ecosystem metabolism (NEM) for the detritus canopy for the three measurement campaigns (g, h, i). Seabed PAR and $O_2$ fluxes are shown as mean ± 1 s.d. and are binned by the hour of day.

There was a significant positive relationship between near-bed incident PAR and the benthic $O_2$ flux (Fig. 5). Light-saturation curves fitted to hourly data from all deployments indicated a maximum gross primary production rate ($P_m$) of 5.14 ± 0.56 mmol $O_2$ m$^{-2}$ h$^{-1}$, an $\alpha$ of 0.03 ± 0.01, and a R rate of 1.92 ± 0.26 mmol $O_2$ m$^{-2}$ h$^{-1}$. Light saturation ($I_k$) of the detritus canopy occurred at irradiances greater than ~170 μmol PAR m$^{-2}$ s$^{-1}$.

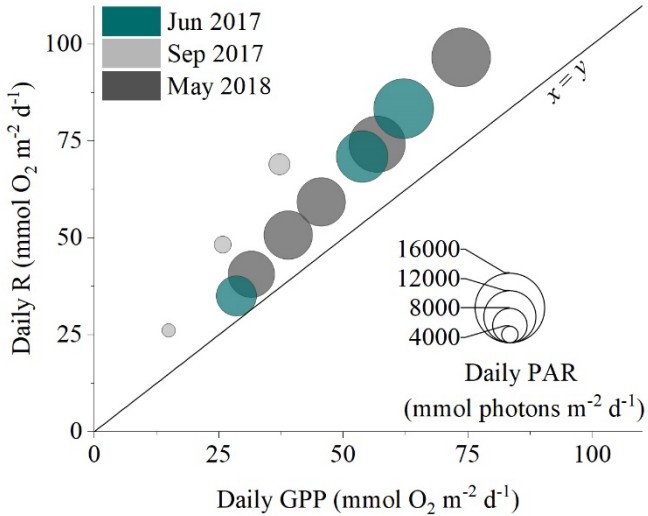

323

Fig. 5: The daily balance between detritus gross primary production (GPP) and respirati®(R) for the
three measurement campaigns. Symbol size corresponds to the daily integrated PAR reaching the
seafloor.

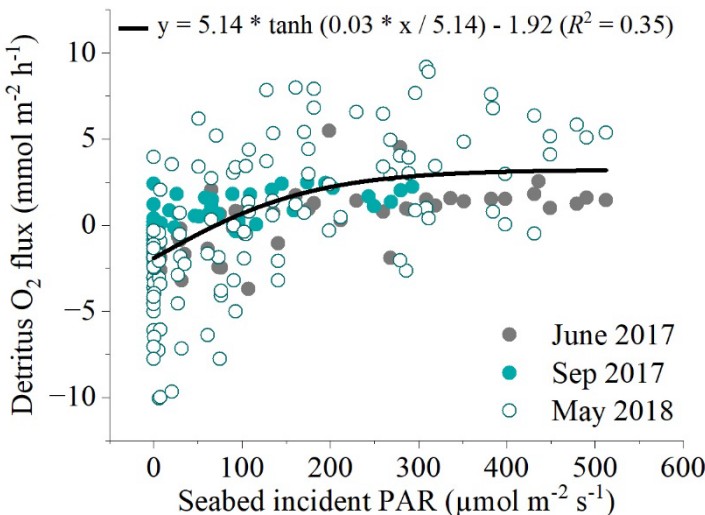

327

Fig. 6: Relationship between all hourly in situ benthic $O_2$ fluxes at the detritus site and light
availability from the three flux datasets measured. A modified photosynthesis-irradiance curve by
Platt et al. (1980) is shown as the line-of-best-fit to the global dataset.

*3.5. Macrobenthic diversity and abundance*

The detritus site had a biomass of accumulated macrophyte (*F. vesiculosus*) detritus of 1666 ± 223
g dry weight $m^{-2}$ (mean ± SE, $n = 4$), approximately 1.5-fold larger than nearby intact *F. vesiculosus*

canopies (Table 2). Detritus accumulation in the five other habitats was around 100-fold smaller. The area of the detritus site estimated using Google Earth was 2300 m$^2$, amounting to 3,800 kg dry weight of *F. vesiculosus* fragments. Macrofauna abundance ranged from 2700 ± 900 ind. m$^{-2}$ at the bare sediments site to 17300 ± 2400 ind. m$^{-2}$ at the sheltered *F. vesiculosus* site (mean ± SE, *n* = 4) (Table 3). Macrofauna biomass ranged from 6 ± 2 g m$^{-2}$ at the bare site to 41 ± 9 g m$^{-2}$ at the exposed seagrass site (mean ± SE, *n* = 4), and the number of species ranged from 6 to 23, with the lowest values measured at the bare sediments and detritus sites, and the highest values at the sheltered *F. vesiculosus* site (Table 3).

At the detritus site, there was a low abundance of epifaunal crustaceans when compared to other habitats with canopies. Key species, such as the amphipod *Gammarus spp.* were notably absent, and isopods such as *Idotea spp.* were present in low abundance (Table A3). Similarly, there was a notable absence of bivalves such as the soft-shelled clam, *Mya arenaria*, and the cockle *Cerastoderma glaucum*. Polychaetes such as *Hediste diversicolor* and *Marenzelleria spp.* were also absent from the detritus site but present in other sedimentary habitats (Table A3). The nMDS ordination of the macrofaunal assemblages indicated a clear separation of points representing the different habitat sites (ANOSIM: $R^2$ = 0.865; p < 0.001). The assemblages from the bare sand and the detritus sites formed separated site groupings compared to the vegetated sites ('*Fucus*' and 'seagrass', both exposed and sheltered). Within the vegetated sites, the assemblages of the 'seagrass sheltered' and the '*Fucus* sheltered' sites were the most different (Fig. 7).

Table 2: Vegetation abundance and biomass (dry weight) at the six study sites. Abundance is shoots per m$^2$ for seagrass and individuals per m$^2$ for *F. vesiculosus*. Values are mean ± SE.

| Site | Abundance per m$^2$ | Above-ground biomass (g m$^{-2}$) | Belowground biomass (g m$^{-2}$) | Detritus (g m$^{-2}$) | Biomass other species (g m$^{-2}$) |
|---|---|---|---|---|---|
| Macrophyte detritus | - | - | - | 1666 ± 223 | - |
| Bare sediments | - | - | - | - | - |
| Sheltered *Z. marina* | 768 ± 92 | 21 ± 2 | 8 ± 1 | 58 ± 13 | 0.1 ± 0.1 |
| Exposed *Z. marina* | 2565 ± 164 | 69 ± 7 | 25 ± 3 | 16 ± 2 | 0.2 ± 0.2 |
| Sheltered *F. vesiculosus* | 16 ± 2 | 1244 ± 58 | - | 55 ± 11 | - |
| Exposed *F. vesiculosus* | 16 ± 2 | 1112 ± 119 | - | 20 ± 2 | - |

Table 3: Macrofauna abundance, biomass (ash-free dry weight), and number of species at the six study sites.

| Site | Infauna abundance (ind. m$^{-2}$) | Epifauna abundance (ind. m$^{-2}$) | Total abundance (ind. m$^{-2}$) | Infauna biomass (g m$^{-2}$) | Epifauna biomass (g m$^{-2}$) | Total biomass (g m$^{-2}$) | Number of species |
|---|---|---|---|---|---|---|---|
| Macrophyte detritus | 4175 ± 2885 | 493 ± 37 | 4668 ± 2885 | 5 ± 3 | 5 ± 0 | 9 ± 3 | 6 |
| Bare sediments | 2719 ± 854 | - | 2719 ± 854 | 6 ± 2 | - | 6 ± 2 | 6 |
| Sheltered *Z. marina* | 6110 ± 787 | 3020 ± 874 | 9130 ± 1176 | 30 ± 6 | 2 ± 0 | 33 ± 6 | 18 |
| Exposed *Z. marina* | 6959 ± 620 | 3316 ± 772 | 10275 ± 990 | 31 ± 8 | 10 ± 2 | 41 ± 9 | 16 |
| Sheltered *F. vesiculosus* | - | 17259 ± 2421 | 17259 ± 2421 | - | 11 ± 2 | 11 ± 2 | 23 |
| Exposed *F. vesiculosus* | - | 3551 ± 609 | 3551 ± 609 | - | 7 ± 2 | 7 ± 2 | 12 |

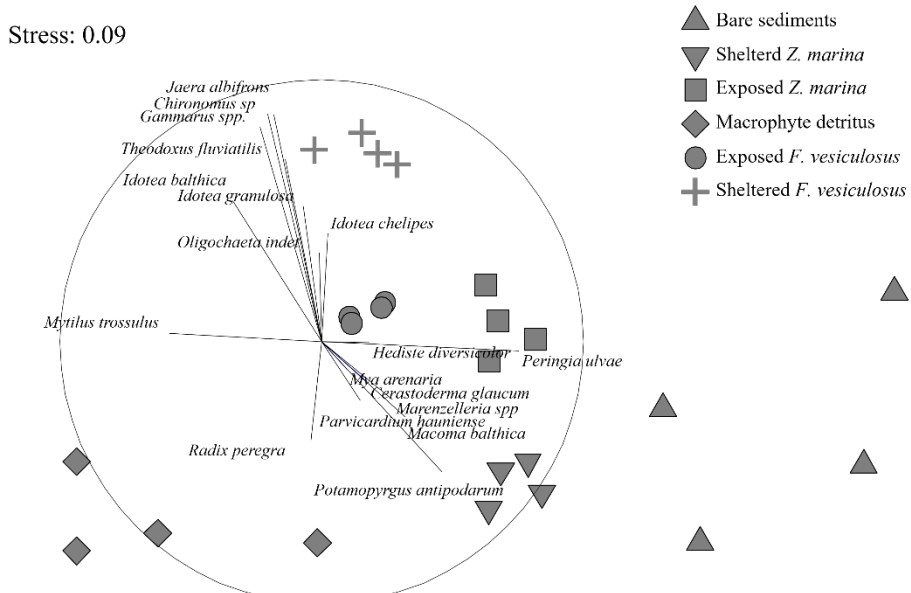

Fig. 7: A non-metric multidimensional scaling (nMDS) ordination of the macrofaunal assemblages indicated a clear separation of points representing the different habitat sites. The assemblages from the bare sand and the detritus sites formed separate site groupings compared to the vegetated sites. Data from May/June 2018 (see Table 1).

## 4. Discussion

### 4.1. Detritus metabolism

The eddy covariance measurements document a highly active detrital canopy that photosynthesized as well as respired. High daily rates of GPP of up to 75 mmol $O_2$ m$^{-2}$ d$^{-1}$ and R of 100 mmol $O_2$ m$^{-2}$ d$^{-1}$ are comparable to some of the most productive habitats in the area, such as dense seagrass meadows (*Zostera marina*) and intact canopies of bladder wrack (*Fucus vesiculosus*) (Attard et al., 2019b). However, intact canopies of *F. vesiculosus* function very differently to detritus canopies from a metabolic standpoint. In June 2017, two eddy covariance instruments were deployed in parallel: one at the detritus site, and another at a nearby intact canopy. While the detritus was net heterotrophic (NEM = -15 mmol $O_2$ m$^{-2}$ d$^{-1}$; GPP:R = 0.76), the intact *F. vesiculosus* canopy was strongly net autotrophic (NEM = 167 mmol $O_2$ m$^{-2}$ d$^{-1}$; GPP:R = 6.40) (Attard et al., 2019b). Daily R at the detritus site was up to ~5-fold larger than that at a nearby (within 4 km) site with bare sediments and up to twice as high as a neighbouring intact canopy of *F. vesiculosus* (Attard et al., 2019b). Decaying (and respiring) fragments of *F. vesiculosus* could contribute substantially to the $O_2$ uptake rate: laboratory incubations of *F. vesiculosus* fragments resolved respiration rates ~5

μmol $O_2$ g dw$^{-1}$ h$^{-1}$, equivalent to ~25 mmol $O_2$ m$^{-2}$ d$^{-1}$ when upscaled to *in situ* biomass observed at the detritus site (data not shown).

Notwithstanding the key metabolic differences between detritus and other neighbouring sites, the flux measurements (Figure 4) indicate that shallow detritus accumulation zones are not just regions of organic matter remineralization, but rather they synthesize substantial amounts of organic matter through primary production. The range in daily GPP:R from 0.53 to 0.81 indicates that primary production can offset a substantial proportion of the respiratory demand, which extends the persistence of detritus in the coastal zone. These observations are consistent with the laboratory study by Frontier et al. (2021), who determined that following detachment, kelp (*Laminaria hyperborea* and *L. ochroleuca*) fragments retain physiological and reproductive capabilities for up to several months. Carbon retention within the coastal zone and export to deeper, sedimentary accumulation regions would therefore be larger than would be predicted by decomposition theory alone. Similarly, slow, and incomplete degradation of algae detritus under low [$O_2$] conditions, which could occur, for instance, in the bottom layers of detrital canopies or in the large anoxic basins of the Baltic Sea (Conley et al., 2009), would increase carbon retention, transfer, and sequestration potential (Pedersen et al., 2021).

*4.2. Periodic benthic hypoxia*

Our in situ measurements performed over a few days in late spring document that subtidal detritus accumulation zones uniquely experience dynamic [$O_2$] conditions driven by sunlight availability and flow velocity, with rapid [$O_2$] oscillations and frequent periods of hypoxia (Table 1). Hypoxic conditions were largely restricted to the lower ~5 cm of the canopy and were only revealed by sensors placed directly above the sediment surface (< 5 cm distance). At the onset of wave-driven mixing, hypoxic waters from within the canopy propagated upwards into the water column and were registered by fast-response [$O_2$] sensors located 10 cm above the canopy (~35 cm above the seafloor). This observation suggests that the [$O_2$] conditions inside the entire canopy and even in the water column directly above can reach hypoxic conditions for a few minutes (Fig. 3). Such pulses, however, were not registered by the slow-response [$O_2$] optodes with a factory-specified T$_{90}$ < 2 min. The minimum [$O_2$] observed by these sensors placed at 10 cm and 35 cm above the seafloor was 158 and 229 μmol L$^{-1}$, respectively, and thus well above hypoxic conditions.

The importance of measuring [$O_2$] close to the seafloor was demonstrated more than 40 years ago by Jorgensen (1980), and since then, other researchers have investigated the distribution of

dissolved constituents such as $O_2$ and nutrients in the benthic boundary layer (Holtappels et al.,
2011). These studies document that solute gradients are largest near the seafloor. For practical
reasons, however, coastal monitoring programs measure $[O_2]$ further away from the seafloor.
Models based on monitoring data suggest that hypoxia is prevalent in only small areas of the
shallow subtidal zone. For instance, models for the northern Baltic Sea, which cover a total seabed
area of 12435 $km^2$ of which 2211 $km^2$ is in shallow waters <5 m depth, indicate that just 16.5 $km^2$
(or 0.75% of shallow waters) are prone to hypoxia (Virtanen et al., 2019). Given that large
quantities of drifting macrophytes are a common phenomenon in the shallow subtidal zone of the
northern Baltic Sea (Norkko and Bonsdorff, 1996a), it is likely that coastal hypoxia is currently
underestimated because large-scale models are largely based on measurements performed higher
above the seafloor (0.5-1.0 m) (Virtanen et al., 2019; Conley et al., 2011).
*4.3. Biodiversity and $[O_2]$ dynamics in detritus canopies*
Despite being considered a temporary habitat, detritus was found in abundance at our study site on
all occasions in May, June, and September. This type of habitat is likely quite widespread in the
Baltic. Habitat distribution models for the area indicate a dominance of *F. vesiculosus* canopies in
shallow waters < 5 m depth (Virtanen et al., 2018), and these canopies are expected to export
substantial amounts of organic matter (~0.3 kg C $m^{-2}$ $yr^{-1}$) which can accumulate in topographical
depressions with limited water exchange (Attard et al., 2019a). Topographic depressions occupy
~1350 $km^2$ or ~11% of the northern Baltic Sea (Virtanen et al., 2019). During a recent seasonal
study, we observed the highest abundance of detritus at our study site in summer and autumn,
coinciding with high southerly winds that erode intact canopies in shallower waters (Attard et al.,
2019a). However, we also observed significant canopy erosion in winter when a substantial biomass
of *F. vesiculosus* froze into sea ice and got dislodged once the ice broke up (Fig. 7). Therefore,
some degree of drifting detritus might be common throughout the year. Drifting detritus constitutes
a significant habitat structure. Given high enough biomass, however, detritus canopies can be a
challenging habitat for most species. At our study site, hypoxic conditions uniquely occurred at the
detritus site and for around a quarter of the deployment time (Table 1). We can expect these
conditions to be particularly challenging for crustaceans, the most hypoxia-sensitive
macroinvertebrate group (Vaquer-Sunyer and Duarte, 2008). Indeed, we only found one crustacean
species at this site- the isopod *Idotea balthica* (Table A3)- which is mobile and can tolerate hypoxic
conditions for a few hours (Vetter and Dayton, 1999). All other invertebrates observed at the
detritus site were mollusks (Table A3), the most hypoxia-tolerant marine invertebrate group
(Vaquer-Sunyer and Duarte, 2008). Other tolerant species include the blue mussel *Mytilus trossulus*
*x edulis* that can survive > 300 h of anoxia (Jorgensen, 1980), although the survival of larvae
depends on its developmental stage (Diaz and Rosenberg, 1995). Similarly, the mudsnail
*Peringia ulvae* is highly mobile and can survive > 150 h of anoxia (Jorgensen, 1980; Norkko et al.,

448    2000).

Overall, the dynamic [$O_2$] conditions in detrital canopies seem to be challenging for most species in
this region of the Baltic Sea, with lethal and non-lethal thresholds frequently being exceeded on
timescales of hours to days. We currently have a poor understanding of the extent of periodic
hypoxia in coastal waters, because [$O_2$] measurements are performed at some distance away from
the seabed. While this is a practical approach that is done to minimize sensor fouling and damage, it
does not reveal the full extent of coastal hypoxia. If implemented widely, sensor arrays, as
described herein, and sensor elevators (e.g. (Holtappels et al., 2011)) can fill in this knowledge gap
and provide important insights into the ecological status and biogeochemical cycling that is needed
for the sustainable management of coastal ecosystems.

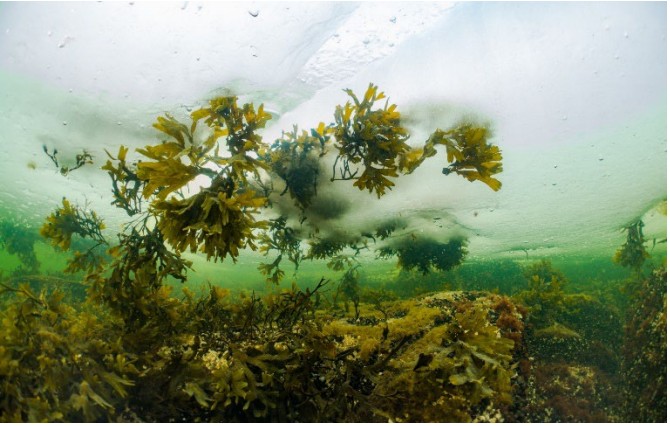


Fig. 8: substantial detritus accumulation was observed in late winter (March 2021) when *F.*
*vesiculosus* froze into sea ice and got dislodged once the ice broke up. (Photo by Alf Norkko)
Table A1: A summary of the eddy covariance flux measurements performed on the detritus canopy
during the three measurement campaigns. Daily integrated seabed PAR and detritus light-use
efficiency (LUE, calculated as daily GPP/ daily PAR) are also presented.

| Field campaign | Day | Daily GPP (mmol $O_2$ $m^{-2}$ $d^{-1}$) | Daily R (mmol $O_2$ $m^{-2}$ $d^{-1}$) | GPP:R | Daily PAR (mmol photons $m^{-2}$ $d^{-1}$) | LUE ($O_2$ photon$^{-1}$) |
|---|---|---|---|---|---|---|
| Jun 2017 | 1 | 62 | 83 | 0.74 | 13554 | 0.005 |
| | 2 | 54 | 71 | 0.76 | 11710 | 0.005 |
| | 3 | 29 | 35 | 0.81 | 9044 | 0.003 |
| Sep 2017 | 1 | 15 | 26 | 0.57 | 3013 | 0.005 |
| | 2 | 37 | 69 | 0.54 | 4827 | 0.008 |
| | 3 | 26 | 48 | 0.53 | 3815 | 0.007 |
| May 2018 | 1 | 46 | 59 | 0.77 | 10997 | 0.004 |
| | 2 | 57 | 74 | 0.76 | 12732 | 0.004 |
| | 3 | 74 | 97 | 0.76 | 13336 | 0.006 |
| | 4 | 32 | 41 | 0.78 | 10523 | 0.003 |
| | 5 | 39 | 51 | 0.77 | 10915 | 0.004 |



Table A2: Fit statistics for linear regressions performed between daily detritus GPP and R, and daily
GPP and benthic PAR. Where relevant, values are presented ± SE. The SE was scaled with the
square root of the reduced Chi-Sqr. ANOVA was used to test slope significance. Asterisks indicate
that the slope was significantly different from zero at the 0.05 level.

| Relationship between daily GPP and daily R | | | | |
|---|---|---|---|---|
| Field campaign | Slope of linear regression ± SE | Intercept ± SE | $R^2$ | ANOVA Prob > F |
| Jun 2017 | 1.43 ± 0.02 | -5.91 ± 0.77 | 0.99 | 0.01* |
| Sep 2017 | 1.93 ± 0.06 | -2.19 ± 1.70 | 0.99 | 0.02* |
| May 2018 | 1.33 ± 0.00 | -1.09 ± 0.17 | 0.99 | 0.00* |
| Global | 1.16 ± 0.13 | 9.90 ± 5.92 | 0.89 | 0.00* |
| | | | | |
| Relationship between daily GPP and daily PAR | | | | |
| Field campaign | Slope of linear regression ± SE | Intercept ± SE | $R^2$ | ANOVA Prob > F |
| Jun 2017 | 128 ± 23 | 5293 ± 1164 | 0.94 | 0.11 |
| Sep 2017 | 82 ± 4 | 1765 ± 121 | 0.99 | 0.03* |
| May 2018 | 73 ± 12 | 8103 ± 609 | 0.90 | 0.01* |
| Global | 182 ± 40 | 1725 ± 1852 | 0.66 | 0.00* |



Table A3: Species list for the five studied sites. Presence is indicated by 'x'.

| Group | Species | Macrophyte detritus | Bare sediments | Sheltered *Z. marina* | Exposed *Z. marina* | Sheltered *F. vesiculosus* | Exposed *F. vesiculosus* |
|---|---|---|---|---|---|---|---|
| Crustacea | *Amphibalanus improvisus* | | | x | | | |
| | *Asellus aquaticus* | | | | | x | |
| | *Corophium* spp. | | | x | | | |
| | *Gammarus* spp. | | | x | x | x | x |
| | *Idotea balthica* | x | | | x | x | x |
| | *Idotea chelipes* | | | | x | x | x |
| | *Idotea granulosa* | | | x | x | x | x |
| | *Jaera albifrons* | | | x | x | x | x |
| | Cladocera | | | | | x | |
| | Copepoda | | | | | x | |
| | Ostracoda sp. | | | | | x | |
| | Mysid | | | | | x | x |
| Bivalvia | *Cerastoderma glaucum* | | | x | x | | |
| | *Parvicardium haunience* | | | x | x | | |
| | *Macoma balthica* | x | x | x | x | x | x |
| | *Mya arenaria* | | | x | x | | |
| | *Mytilus trossulus x edulis* | x | | x | x | x | x |
| Gastropoda | *Peringia ulvae* | x | x | x | x | x | x |
| | *Radix* sp. | x | | x | | | x |
| | *Potamopyrgus antipodarum* | | x | x | | | |
| | *Theodoxus fluviatilis* | x | x | x | x | x | x |

| | | | | | | | |
|---|---|---|---|---|---|---|---|
| Polychaeta | *Hediste diversicolor* | | | x | x | | |
| | *Halicryptus spinulosus* | | | | | x | |
| | *Maranzelleria* spp. | | x | x | x | x | |
| | Nematoda | | | | | x | |
| | Oligochaeta | | | x | x | x | |
| | *Pygospio elegans* | | | | | x | |
| Others | *Chironomus* sp | | | x | x | x | x |
| | Coleoptera larvae | | | | | | x |
| | Odonata | | | | | | x |
| | *Cyanophthalma obscura* | | | | | | x |
| | Hydrachnidae | | x | | | | x |


## Author contribution

All authors contributed significantly to designing the research, funding the study, collecting the data, analyzing samples and data, and interpreting the results. KMA wrote the paper with input from all authors.

## Competing interests

The authors declare that they have no conflict of interest

## Data availability

All data presented in this paper will be made available in a FAIR-aligned data repository upon acceptance of the paper.

## Acknowledgements

Colleagues at the Tvärminne Zoological Station provided help with fieldwork and logistics. Anni Glud at the University of Southern Denmark constructed the oxygen microsensors used in this study. Elina Virtanen at the Finnish Environmental Institute (SYKE) provided spatial data used to estimate the potential extent of detritus canopies. The Walter and Andrée de Nottbeck Foundation supported this work through a postdoctoral fellowship to KMA and through a Masters fellowship to AL. Further funding for this project was provided by research grants from the Academy of Finland (project ID 294853), the University of Helsinki and Stockholm University strategic fund for collaborative research (the Baltic Bridge initiative), and Denmark's Independent Research Fund (project ID 7014-00078). This study has utilized research infrastructure facilities provided by FINMARI (Finnish Marine Research Infrastructure network, The Academy of Finland, project ID 283417).

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
