# Peer review of "Title: Drifting macrophyte detritus triggers 'hidden' benthic hypoxia"

_Biogeosciences, 2022_

## Referee Comment (RC2)

**General comments**

The manuscript "D*rifting macrophyte detritus triggers "hidden" benthic hypoxia*" investigates how a detritus mat of macroalgae affects oxygen conditions along the benthos of the Baltic Sea. The authors put their observations in context of other benthic habitats in their study area. The authors also investigate the metabolism of the detritus mats at three separate occasions (2 seasons). They find that hypoxia in the bottom layer of detritus aggregations occurs whenever water velocity is low (ca. 2 cm $s^{-1}$), with reoxygenation of the mat happening at ca. 7 cm $s^{-1}$. The authors then link measurements of mat metabolism with the observed fluctuations in oxygen.

I would personally like to apologise to the authors for my very late review, as caused by some personal circumstances.

The manuscript is clear, concise and impeccably written. My principal criticism is that, at present, the manuscript does not a good job at establishing its scientific novelty and discussing the relevance of its findings within the context of hypoxia in the Baltic sea (a well-documented and important phenomenon). The effects of macrophyte detritus mats on oxygen concentrations and benthic fauna are well documented (e.g. (Tzetlin et al. 1997; Mascart et al. 2015; Hendy et al. 2021)), including in the Baltic Sea (e.g. (Sundbäck et al. 1989; Bonsdorff 1992; Norkko et al. 2000; Berezina 2008), so the manuscript needs to do a better job in clearly outlining its scientific contribution. The authors have a nice dataset of high-resolution measurements, which offers the opportunity to move towards a more mechanistic—albeit correlative—understanding of the drivers of hypoxia in macroalgal accumulations and shallow benthos. While the graphs presented clearly show a relationship between flow velocity and light availability, a more formal analysis of the data (even if it is just a correlation analysis cf. Fig. 6) would improve the reader's confidence.

That is important as there could be other (unmeasured) drivers that may be somewhat influencing the oxygen concentration. For instance, Fig. 2 shows no hypoxia towards the morning of Day 3 (end of the graph) despite a ~3 hr period of slow water velocity, which contrasts with the really rapid development of hypoxia in Day 3 as soon as water velocity slows. Similarly, there is no rapid recovery period (cf. Fig. 3) at night on Day 2 despite high flow. Is that related to the light conditions? Hard to tell without a more robust inspection of the relationships.

[Figure]

In that context, I missed a more formal discussion of influence of sediment metabolism, salinity and the halocline on the observations, given that they are known to be important drivers of the oxygen dynamics in the Baltic. For instance, the authors also took high-resolution measurements on a nearby (~4km) sediment community, so not comparing the results with the ones from the detritus aggregation more explicitly seems like a missed opportunity. Such analysis could help the reader better understand how sediment metabolism can influence oxygen in the study area. This is important as it can help solidify the link between detritus metabolism and oxygen fluxes, which is currently not fully developed (see comment below in discussion).

Another area that would benefit from improvement is the contextualization of the results. The Baltic Sea is well-known to be prone to hypoxia, with multiple drivers acting at different spatial scales. A better description of that system in the Introduction would help frame the importance of the study's aims. I suggest writing a paragraph about Baltic hypoxia and the existing knowledge gaps. Additionally, further discussion and contextualization of the results beyond the study area would also improve the manuscript. How prevalent may be *Fucus* detrital aggregations given its cover in the Baltic? What may be their relative importance in driving hypoxia compared to the more well-studied aggregations of filamentous algae?

Overall I was not convinced about the "hidden hypoxia" angle given that this is a well-documented phenomenon as the authors point out (e.g. Jorgensen 1980, see also some of the references I included) and so it is really not "hidden" at all. The reason why we don't measure that hypoxia in monitoring programs is probably practical. I would advise on minimizing that angle in the title, intro and discussion. The bigger contribution on the manuscript is somewhere else, e.g. in the high resolution measurements and examination of oxygen drivers. If the authors decide to continue on the "hidden" hypoxia angle, I would advise on elaborating further on why does it matter that we can detect small-scale hypoxia near the sediment surface.

**Specific comments**

There should be a better distinction between the sections and experiments conducted, as the titles "O2 dynamics" (section 2.2) and "O2 fluxes" (e.g. section 2.3) are a bit confusing. To someone that is not familiar, it may seem unclear why you use oxygen sensor array in one instance and AEC in another. Please outline that better.

I also found it hard to know when each of the measurements were taken, and why some of the results were not included in the figures. For instance why are only 2/3 of the measurements shown in Fig. 4? It may be valid to not include some measurements, but the reader is left wondering why if no explanation is not provided. I suggest all the figures have their date of sampling included to help better guide what set of deployments the reader is looking at (e.g. oxygen array vs AEC).

Ln. 135. Please outline better what is it that you want to measure with this technique and why.

Ln. 163. This is a common assumption, but studies from several systems show that $R_{light}$ may be higher than $R_{dark}$. Do we know how well this assumption prevails in macroalgae systems? Including a reference may help.

Ln. 260. Does that correspond to a daylight or night-time period?

Ln. 270. Personally I felt that manuscript needs to link this finding with the mat measurements better, either here in the results or discussion section.

Ln. 330. How do we know that those O2 fluxes are the result of the detritus canopy and not the photosynthetic community within the sediment? A better case needs to be presented here. Consider including measurements on bare sediment area

Ln. 331. This is however not the main finding of the study. The Discussion would benefit from stating more upfront what the main findings of the study area in a succinct manner. E.g. you observed hypoxia in the bottom of the mat, and you link that to mat metabolic activity combined with water flow.

Ln. 360-370. Personally I found this paragraph a bit out of place. It seems like discussing the consequences of the hypoxia you document in the previous paragraph for faunal communities (which you measured) would flow better here.

Ln. 380. This section is quite confusing, as it is simultaneously talking about detritus agregations, habitat structure (ln. 380), oxygen dynamics (ln. 384) and the effects of detritus mats on diversity (391). I suggest splitting it into different paragraphs. E.g. you can talk about the prevalence (seasonal, spatial) of Fucus detritus in the study area and the Baltic, the consequences of hypoxia for faunal communities, and the consequences of macroalgae-induced hypoxia for sediment communities in different paragraphs, as there is plenty to elaborate there on.

Fig. 4. It would be useful if the panels had the night-time and daytime overlayed on the deployment time axis (e.g. shaded box for night). Please consider doing that for the first panel rows of Fig. 2 and 5 as well. Also please include date of measurements

---

## Author Response (AR1)

**Author response to Reviewer 1: Dr. Dirk Koopmans**

This is valuable work on an under-studied benthic ecosystem. The manuscript presents oxygen fluxes over macrophyte (*F. vesiculosus*) detritus that accumulated in a topographical depression in shallow waters of the Baltic Sea. The manuscript is well-written and concise. The methods are clearly presented and appropriate to the goals. The primary findings are 1) that hypoxia occurs frequently in the depression and periodically in overlying water, and 2) that there is substantial detrital photosynthesis. These findings support the broader implications that 1) benthic hypoxia in shallow waters of the Baltic Sea is underestimated, and 2) the retention and export of *F. vesiculosus* carbon from coastal zones is likely greater than previously estimated.

I enjoyed this work and have minor suggestions for its improvement. My primary criticism is that the manuscript does not provide more context for metabolism of the detritus, nor for its contribution to the occurrence of shallow water hypoxia in the Baltic Sea.

On the first point, it is surprising that detrital gross primary production was so close to respiration, particularly in May and June. The authors begin the Discussion by comparing GPP to that of attached *F. vesiculosus* canopies (line 333), but it would be helpful to provide more detail. I recommend that the authors add a figure that shows how the metabolism of these detrital canopies differs from that of attached *F. vesiculosus* canopies.

Response: We are grateful to Dr. Dirk Koopmans for providing a thoughtful review of our work. To provide a more detailed comparison of detritus metabolism to intact canopies, we propose (1) including an additional panel in Fig. 5 that includes daily GPP, R, and NEM of intact *F. vesiculosus* canopies from our previous work (Attard et al. 2019a), and (2) describing this panel and the comparison explicitly in a few sentences in the Discussion under section 4.1.: Detritus metabolism rates.

Update: We now describe differences between intact and detritus canopies explicitly in the Discussion under section 4.1: Detritus metabolism rates. The text is reproduced below. We decided not to add another panel to Figure 5 because we felt it would make the figure look overly busy with data. **Please note that line numbers refer to the document with 'simple markup'**

L372-382: "However, intact canopies of *F. vesiculosus* function very differently from a metabolic standpoint. In June 2017, two eddy covariance instruments were deployed in parallel: one at the detritus site, and another at a nearby intact canopy. While the detritus was net heterotrophic (NEM = -15 mmol $O_2$ m$^{-2}$ d$^{-1}$; GPP:R = 0.76), the intact *F. vesiculosus* canopy was strongly net autotrophic (NEM = 167 mmol $O_2$ m$^{-2}$ d$^{-1}$; GPP:R = 6.40) (Attard et al., 2019b). Daily R at the detritus site was up to ~5-fold larger than that at a nearby (within 4 km) site with bare sediments and up to twice as high as a neighbouring intact canopy of *F. vesiculosus* (Attard et al., 2019b). Decaying (and respiring) fragments of *F. vesiculosus* could contribute substantially to the $O_2$ uptake rate: laboratory incubations of *F. vesiculosus* fragments resolved respiration rates ~5 μmol $O_2$ g dw$^{-1}$ h$^{-1}$, equivalent to ~25 mmol $O_2$ m$^{-2}$ d$^{-1}$ when upscaled to *in situ* biomass observed at the detritus site (data not shown). Notwithstanding these key differences, the flux measurements (Figure 5) indicate that shallow detritus accumulation zones are not just regions of organic matter remineralization, but rather they synthesize substantial amounts of organic matter through primary production."

On the second point, I recommend that the authors provide more perspective on the occurrence of shallow-water hypoxia in the Baltic Sea, and how *F. vesiculosus* detritus may contribute to it. Where else is shallow-water hypoxia observed? Does it naturally occur elsewhere (apart from areas of high anthropogenic impact)? Based on prior work, can one estimate how much detritus is exported from attached *F. vesiculosus* per year? Given this export and your results, what area of the topographical depressions in shallow water of the Baltic could behave as you have observed here? I acknowledge that there are complicating factors which may prohibit the authors from estimating this. For example, much of *F. vesiculosus* detritus decomposes in the intertidal zone and therefore would not contribute to oxygen uptake in shallow depressions. Nevertheless, it would be valuable to include a discussion of knowns and unknowns.

**Response**

**Q1: Where else is shallow-water hypoxia observed?** Information on shallow-water hypoxia is generally scarce, but we have some numbers that we will include in the revision. Our key reference is the study by Virtanen et al. (2019) for the northern Baltic Sea (Gulf of Finland and Archipelago Sea). This region has a total seabed area of 12435 km2 and a shallow-water area (0-5 m depth) of 2211 km2. Based on their model, the total area prone to hypoxia is 1351 km2 (all depths) and 16.5 km2 for shallow areas < 5 m depth. Of the 461 monitoring stations in this area of the Baltic Sea that registered hypoxia, only 11 were in waters < 5 m depth. These are likely underestimates since the O2 measurements driving the models are done 1m above the seafloor. We will state this explicitly in the revision.

Update: we now include a section in the introduction highlighting the results of Virtanen et al. (2019) more explicitly:

L68-75: "To date, records of hypoxia in the shallow subtidal zone are still somewhat scarce. In a compilation of monitoring data for the northern Baltic Sea (Gulf of Finland and Archipelago Sea), Virtanen et al. (2019) found that just 11 out of 461 (or 2.4%) of the monitoring stations that registered hypoxia occurred in waters < 5 m depth. While this may reflect a true signal that hypoxia is more widespread in deeper coastal waters, it is also likely that hypoxic conditions go undetected if measurements are performed away from the seafloor, as is common practice (Conley et al., 2011; Virtanen et al., 2019)."

We also include a new section in the Discussion:

L415-422: "Models based on monitoring data suggest that hypoxia is prevalent in only small areas of the shallow subtidal zone. For instance, models for the northern Baltic Sea, which cover a total seabed area of 12435 km$^2$ of which 2211 km$^2$ is in shallow waters <5 m depth, indicate that just 16.5 km$^2$ (or 0.75% of shallow waters) are prone to hypoxia (Virtanen et al., 2019). Given that large quantities of drifting macrophytes are a common phenomenon in the shallow subtidal zone of the northern Baltic Sea (Norkko and Bonsdorff, 1996b), it is likely that coastal hypoxia is currently underestimated because large-scale models are based on measurements performed higher above the seafloor (0.5-1.0 m) (Virtanen et al., 2019; Conley et al., 2011)."

**Q2: Does it naturally occur elsewhere (apart from areas of high anthropogenic impact)?** Yes, O2 deplete conditions and even sulfidic conditions are often observed in association with

macrophyte detritus, even in remote and pristine environments such as the high Arctic (Glud et al. 2004, cited in L32). We will state this explicitly in the revision.

Update: We now include a section in the Introduction stating this explicitly. The text is reproduced below.

L88-92: "While coastal hypoxia is most commonly associated with eutrophic waters such as the Baltic Sea (Carstensen and Conley, 2019; Conley et al., 2011), hypoxic (and even sulfidic) conditions have been reported in remote and more pristine environments such as the high Arctic due to large accumulations of detritus produced from perennial brown seaweeds (Glud et al., 2004)."

**Q3: Based on prior work, can one estimate how much detritus is exported from attached *F. vesiculosus* per year? Given this export and your results, what area of the topographical depressions in shallow water of the Baltic could behave as you have observed here?** In a previous study we estimated that *F. vesiculosus* export ~0.3 kg C m-2 yr-1 (Attard et al. 2019b). Given that habitat distribution models for the area indicate a dominance of *F. vesiculosus* in shallow waters < 5 m depth (Virtanen et al. 2018), we have reason to believe that other topographical depressions accumulate macrophyte detritus and would likely function in a similar manner to our study site. We will state this explicitly in the revision.

Update: We now include a section in the Discussion stating this explicitly. The text is reproduced below.

L425-430: "This type of habitat is likely quite widespread in the Baltic. Habitat distribution models for the area indicate a dominance of *F. vesiculosus* canopies in shallow waters < 5 m depth (Virtanen et al., 2018), and these canopies are expected to export substantial amounts of organic matter (~0.3 kg C $m^{-2}$ $yr^{-1}$) (Attard et al., 2019a) which can accumulate in topographical depressions with limited water exchange. Topographic depressions occupy ~1350 $km^2$ or ~11% of the northern Baltic Sea (Virtanen et al., 2019)."

**Minor points**

line 87 - two citations are used. One is relevant to the first half of the sentence, the other is relevant to the second half. I recommend separating the citations to denote the portions of the sentence that they are relevant to.

Response: OK, we will adjust accordingly.

Update: This was implemented as suggested.

line 137 - deployments were performed on June 2017, September 2017, and May 2018, but in Figure 5 the deployments are listed as June 2017, September 2017, and June 2018.

Response: Thank you for catching this inconsistency. The deployment started on 30th May and ran into June. We will adjust to say May 2018.

Update: The figure was updated as suggested.

line 139 - McGinnis instead of Mcginnis.

Response: OK, we will adjust accordingly.

Update: The citation was updated as suggested.

line 152 - "The storage correction term was defined as an average of the O2 sensors located within and above the canopy." Figure 2(b) shows a dissolved oxygen profile within and above the canopy that is not simply an average of the O2 sensors. Why not use this approach to also correct for storage?

Response: This is a good suggestion; we will implement accordingly.

Update: We implemented this storage correction term as suggested. The text is reproduced below.

L165-178: "The storage correction term was defined as a matrix with the number of rows $n$ corresponding to the sensor measurement height above the seafloor (1 row per cm) (Camillini et al., 2021). To do this, the oxygen time series, consisting of $[O_2]$ measurements performed at three heights within the canopy, were converted to a matrix using the software package OriginPro 2022. Since the measurement height of the three sensors were spaced nonlinearly, the data were first converted to XYZ column format using the w2xyz function. Next, the three rows, representing the $[O_2]$ time series measurements at three heights, were expanded to $n$ rows, with $n$ representing the sensor measurement height in cm (from 0 to $n$ cm above seabed, 1 row per cm) using the XYZ Gridding function. This generated a matrix of $n$ rows consisting of linearly interpolated $[O_2]$. Interpolation was performed using the Random (Renka Cline) gridding method. Next, a storage correction term was calculated for each 1 cm cell as described by Rheuban et al. (2014), and the total storage correction was subsequently computed for the water volume below the sensor measurement height as the sum of the $n$ rows."

lines 194-197 - Seagrass leaf length and canopy density were determined, but don't appear again in the results. Perhaps these analyses can be left out of the manuscript.

Response: It is correct that seagrass leaf length does not appear. We will exclude. Seagrass density does however appear in Table 2 (Abundance per m2).

Update: Seagrass leaf length was removed as suggested.

line 209-210. "The wet weight for each species was noted with 0.0001 g accuracy" is an unnecessary statement. I suggest removing it.

Response: It is standard procedure to report accuracy when measuring the biomass of individual animals, which is reported in Table 3. We would like to keep this if possible.

line 237. The sentence begins "In the upper canopy region...", but the preceding sentence already focuses on the upper layers of the canopy. I recommend replacing the quoted words with "There."

Response: OK, we will adjust accordingly.

Update: This was adjusted as suggested.

line 265 - the deployment months are listed here as June 2017, September 2017, and June 2018.

Response: Thanks for catching this. Here and throughout the ms we will adjust to May 2018.

Update: Here and throughout the MS, June 2018 is changed to say May 2018.

Figure 3. The symbol key lists Flow velocity 0.125 s (cm s-1) and Flow velocity, 10 s (cm s-1). I suggest that the word "Mean" be included for the second label.

Response: OK, we will adjust accordingly.

Update: This was done as suggested.

Figure 4. Consider including PAR and rearranging the panels. Panel a could be PAR, panel b could be O2 flux and flow velocity, then panels c and d could be the insets of O2 flux over time.

Response: We will consider including PAR and we will rearrange panels for improved readability.

Update: Following reviewer comments highlighted above and below, we decided to exclude Figure 4 since we do not think that this figure is required.

Figure 4. It is not clear that the insets provide valuable data to the manuscript. I suggest either referencing those rates explicitly in the manuscript or removing them.

Response: The insets are meant to illustrate the increase in [O2] slope quantitatively by including a regression. We will add PAR as suggested above and evaluate whether these panels are still required.

Update: We decided to exclude Figure 4 since we do not think this figure is required.

Figure 5. Again consider including PAR in the figure. It is useful to compare across seasons and to align with observed changes in flux.

Response: We will consider including PAR although we need to evaluate whether the figure will become overly busy with data.

Update: We added PAR to this figure. The new figure is reproduced below. We updated the figure legend accordingly.

[Figure]

Figure 6. Consider coloring the O2 flux symbols by the time of year.

Response: OK, we will color the symbols to reflect the three different field campaigns.

Update: We now show the O2 flux symbols mapped by the time of the year. The new image is shown below.

[Figure]

line 299 - the number of significant digits is inconsistent in this section. "...area of detritus was 2300 m2, amounting to 3,832 kg dry weight..." I suggest rounding all numbers, including microfaunal abundance, to an appropriate significant digit e.g., 17259 to 17300.

Response: OK, we will round figures as suggested.

Update: This was implemented as suggested.

line 346 - It would be useful to include a short analysis of factors that could use seasonal differences in GPP and R in the detrital canopy. GPP in particular appears smaller in September than in the earlier months.

Response: We will add a small table summarizing the eddy covariance deployments (duration, daily integrated seafloor PAR, water temperature). There is a clear coupling between daily GPP and R (Fig. 5) which is likely driven by changes in daily PAR. We will include an analysis of this in the revised manuscript.

Update: We added a new analysis including a new figure 6, reproduced below. We added new text to the Results section. We added a new table in appendix A1 summarizing the eddy covariance measurements. We added a new table in appendix A2 summarizing the regression analyses.

L305-311: 'There was a significant positive relationship between daily detritus GPP and R in all measurement campaigns, with the detritus canopy seemingly becoming more heterotrophic (i.e. R > GPP) as the magnitude of the metabolic rates increased (Fig. 5, Table A2). Significant positive relationships were also observed between daily detritus GPP and daily seabed PAR (Table A2). Canopy light-use efficiency (LUE), estimated as the ratio between daily GPP and daily PAR (Attard and Glud, 2020), was 0.004 $O_2$ photon$^{-1}$ in June 2017, 0.006 $O_2$ photon$^{-1}$ in September 2017, and 0.004 $O_2$ photon$^{-1}$ in May 2018 (Table A1).'

line 356 - I believe that the reference to Figure 4 is intended to refer to Figure 3.

Response: Thanks for catching this. Indeed, this should reference figure 3. Will correct.

Update: this was corrected.

line 374 - "Topographical depressions with limited water exchange occupy ~1350 km2 of the northern Baltic Sea" This is interesting but vague. Could the authors provide more details? What is the extent of these depressions relative to surroundings? Are these all in shallow areas? Were there other important characteristics? How was the total area quantified?

Response: As outlined in response to the major comment above, we will include more information regarding extent of hypoxic areas for all depths and for shallow areas in this region of the Baltic Sea. We will also comment on our expectation that other shallow depressions likely function in a similar manner to our study site.

Update: We now include new sections in the Introduction and the Discussion expanding upon these points as highlighted above, as requested.

**Author response to Reviewer 2**

The manuscript "Drifting macrophyte detritus triggers "hidden" benthic hypoxia" investigates how a detritus mat of macroalgae affects oxygen conditions along the benthos of the Baltic Sea. The authors put their observations in context of other benthic habitats in their study area. The authors also investigate the metabolism of the detritus mats at three separate occasions (2 seasons). They find that hypoxia in the bottom layer of detritus aggregations occurs whenever water velocity is low (ca. 2 cm s -1 ), with reoxygenation of the mat happening at ca. 7 cm s-1 . The authors then link measurements of mat metabolism with the observed fluctuations in oxygen.

I would personally like to apologise to the authors for my very late review, as caused by some personal circumstances. The manuscript is clear, concise and impeccably written. My principal criticism is that, at present, the manuscript does not a good job at establishing its scientific novelty and discussing the relevance of its findings within the context of hypoxia in the Baltic sea (a well-documented and important phenomenon).

The effects of macrophyte detritus mats on oxygen concentrations and benthic fauna are well documented (e.g. (Tzetlin et al. 1997; Mascart et al. 2015; Hendy et al. 2021)), including in the Baltic Sea (e.g. (Sundbäck et al. 1989; Bonsdorff 1992; Norkko et al. 2000; Berezina 2008), so the manuscript needs to do a better job in clearly outlining its scientific contribution. The authors have a nice dataset of high-resolution measurements, which offers the opportunity to move towards a more mechanistic—albeit correlative—understanding of the drivers of hypoxia in macroalgal accumulations and shallow benthos.

While the graphs presented clearly show a relationship between flow velocity and light availability, a more formal analysis of the data (even if it is just a correlation analysis cf. Fig. 6) would improve the reader's confidence. That is important as there could be other (unmeasured) drivers that may be somewhat influencing the oxygen concentration. For instance, Fig. 2 shows no hypoxia towards the morning of Day 3 (end of the graph) despite a ~3 hr period of slow water velocity, which contrasts with the really rapid development of hypoxia in Day 3 as soon as water velocity slows. Similarly, there is no rapid recovery period (cf. Fig. 3) at night on Day 2 despite high flow. Is that related to the light conditions? Hard to tell without a more robust inspection of the relationships.

Response: We are grateful to Reviewer 2 for taking the time to provide a thoughtful review of our manuscript. We appreciate that our study scratches the surface of what is a complex topic and that resolving the mechanistic drivers in a causative manner would require a larger multidisciplinary effort. In short, there is much scope for further investigation. We will endeavor to take on reviewer suggestions to improve readability.

Update: To address the point about performing a more formal analysis of the data, we added a new table in appendix A1 summarizing the eddy covariance measurements and we added a new table in appendix A2 summarizing the regression analyses.

Figure 2 is an interesting example and in hindsight, it deserves more words than we allocated in Section 3.2. The reviewer rightly points out that the mean flow velocity alone does not explain the dynamics in buildup/erosion of hypoxia. In fact, from close analysis of the hydrodynamics data, we conclude that it is the prevalence of surface waves, rather than the mean flow velocity, that best explains the buildup of hypoxia. This is mentioned in our abstract (L21: "…hypoxia…terminated at

the onset of wave-driven hydrodynamic mixing") and in the discussion (L352) but unfortunately it isn't elaborated further. Our conclusion is however consistent with direct observations of canopy turbulence and mixing which suggest that surface waves are much more effective at ventilating macrophyte canopies than horizontal flow velocity (Hansen & Reidenbach 2017). Surface waves are evident in the velocity time series in Fig. 2. Looking at the first period with high flow velocity in Day 1, we can see that the variance around the mean flow velocity is quite small, suggesting minimal surface waves. The second period with high velocity in Days 2 and 3 have a much larger variance around the mean, suggesting the presence of significant surface waves. In the revision, we will elaborate on this and include an analysis of wave statistics (wave height, wave orbital velocity) to correlate with the buildup of hypoxia. Our measurements overwhelmingly show that it is the presence of surface waves rather than light or other environmental parameters that determines the buildup of hypoxic conditions in detrital canopies.

Update: we have since caught a mistake in our graph. The time series for flow velocity and [$O_2$] were not properly aligned. We apologize for this. The correct image is shown below. The image shows much more clearly that as the flow velocity increases, the canopy becomes oxygenated and as the flow velocity drops towards the end of the deployment, the [$O_2$] begins to decrease once again. This correction helps to clarify some of the uncertainties that the reviewer rightly mentions. The updated figure is now included in the revised ms.

[Figure]

In that context, I missed a more formal discussion of influence of sediment metabolism, salinity and the halocline on the observations, given that they are known to be important drivers of the oxygen dynamics in the Baltic. For instance, the authors also took high-resolution measurements on a nearby (~4km) sediment community, so not comparing the results with the ones from the detritus aggregation more explicitly seems like a missed opportunity. Such analysis could help the reader better understand how sediment metabolism can influence oxygen in the study area. This is important as it can help solidify the link between detritus metabolism and oxygen fluxes, which is currently not fully developed (see comment below in discussion).

Response: We will include more discussion on the influence of sediment metabolism, in particular the expected O2 consumption driven by the underlying sediments. We quantified this in a previous

study (Attard et al. 2019a). We also performed O2 consumption measurements on *F. vesiculosus* fragments that we could scale up to the biomass to investigate the O2 drawdown- we will include these in the revision. The depth of the halocline is important for seasonal hypoxia but we do not think it will impact the buildup of hypoxia at our shallow site. The halocline typically is located at 30-50m depth.

Update: We now include a section in the Discussion where we compare the habitats explicitly. **Please note that line numbers refer to the document with 'simple markup'**:

L376-382: "Daily R at the detritus site was up to ~5-fold larger than that at a nearby (within 4 km) site with bare sediments and up to twice as high as a neighbouring intact canopy of *F. vesiculosus* (Attard et al., 2019b). Decaying (and respiring) fragments of *F. vesiculosus* could contribute substantially to the $O_2$ uptake rate. Laboratory incubations of *F. vesiculosus* fragments resolved respiration rates ~5 $\mu$mol $O_2$ g dw$^{-1}$ h$^{-1}$, equivalent to ~25 mmol $O_2$ m$^{-2}$ d$^{-1}$ when upscaled to *in situ* biomass observed at the detritus site (data not shown)."

Another area that would benefit from improvement is the contextualization of the results. The Baltic Sea is well-known to be prone to hypoxia, with multiple drivers acting at different spatial scales. A better description of that system in the Introduction would help frame the importance of the study's aims. I suggest writing a paragraph about Baltic hypoxia and the existing knowledge gaps. Additionally, further discussion and contextualization of the results beyond the study area would also improve the manuscript. How prevalent may be Fucus detrital aggregations given its cover in the Baltic? What may be their relative importance in driving hypoxia compared to the more well-studied aggregations of filamentous algae?

Response: We will include a better description of the Baltic Sea and hypoxia, highlighting the knowledge gaps that we aim to tackle with this study (i.e. the prevalence of periodic hypoxia in shallow waters). Reviewer 1 had similar suggestions to elaborate on how widespread shallow-water hypoxia might be. We reproduce our response to Reviewer 1 below.

**Where else is shallow-water hypoxia observed?** Information on shallow-water hypoxia is generally scarce, but we have some numbers that we will include in the revision. Our key reference is the study by Virtanen et al. (2019) for the northern Baltic Sea (Gulf of Finland and Archipelago Sea). This region has a total seabed area of 12435 km2 and a shallow-water area (0-5 m depth) of 2211 km2. Based on their model, the total area prone to hypoxia is 1351 km2 (all depths) and 16.5 km2 for shallow areas < 5 m depth. Of the 461 monitoring stations in this area of the Baltic Sea that registered hypoxia, only 11 were in waters < 5 m depth. These are likely underestimates since the O2 measurements driving the models are done 1m above the seafloor.

**Based on prior work, can one estimate how much detritus is exported from attached *F. vesiculosus* per year? Given this export and your results, what area of the topographical depressions in shallow water of the Baltic could behave as you have observed here?** In a previous study we estimated that *F. vesiculosus* export ~0.3 kg C m-2 yr-1 (Attard et al. 2019b). Given that habitat distribution models for the area indicate a dominance of *F. vesiculosus* in shallow waters < 5 m depth (Virtanen et al. 2018), we have reason to believe that other topographical depressions accumulate macrophyte detritus and would likely function in a similar manner to our study site. We will state this explicitly in the revision. Regarding the impact of filamentous algae, it is difficult to separate out their influence because our study site contained a mixture of F.

vesiculosus and filamentous algae (Fig. 1b), although the detrital biomass was overwhelmingly dominated by F. vesiculosus.

Update: we now include a new section in the Introduction highlighting the knowledge gaps that this study aims to address:

L68-75: "To date, records of hypoxia in the shallow subtidal zone are still somewhat scarce. In a compilation of monitoring data for the northern Baltic Sea (Gulf of Finland and Archipelago Sea), Virtanen et al. (2019) found that just 11 out of 461 (or 2.4%) of the monitoring stations that registered hypoxia occurred in waters < 5 m depth. While this may reflect a true signal that hypoxia is more widespread in deeper coastal waters, it is also likely that hypoxic conditions go undetected if measurements are performed away from the seafloor, as is common practice (Conley et al., 2011; Virtanen et al., 2019)."

We now include a new section in the Discussion stating how widespread we believe this habitat to be in the Baltic. The text is reproduced below.

L425-430: "This type of habitat is likely quite widespread in the Baltic. Habitat distribution models for the area indicate a dominance of *F. vesiculosus* canopies in shallow waters < 5 m depth (Virtanen et al., 2018), and these canopies are expected to export substantial amounts of organic matter (~0.3 kg C $m^{-2}$ $yr^{-1}$) (Attard et al., 2019a) which can accumulate in topographical depressions with limited water exchange. Topographic depressions occupy ~1350 $km^2$ or ~11% of the northern Baltic Sea (Virtanen et al., 2019)."

We also include another section in the Discussion highlighting why we think that coastal hypoxia might be underestimated:

L415-422: "Models based on monitoring data suggest that hypoxia is prevalent in only small areas of the shallow subtidal zone. For instance, models for the northern Baltic Sea, which cover a total seabed area of 12435 $km^2$ of which 2211 $km^2$ is in shallow waters <5 m depth, indicate that just 16.5 $km^2$ (or 0.75% of shallow waters) are prone to hypoxia (Virtanen et al., 2019). Given that large quantities of drifting macrophytes are a common phenomenon in the shallow subtidal zone of the northern Baltic Sea (Norkko and Bonsdorff, 1996b), it is likely that coastal hypoxia is currently underestimated because large-scale models are based on measurements performed higher above the seafloor (0.5-1.0 m) (Virtanen et al., 2019; Conley et al., 2011)."

Overall I was not convinced about the "hidden hypoxia" angle given that this is a well-documented phenomenon as the authors point out (e.g. Jorgensen 1980, see also some of the references I included) and so it is really not "hidden" at all. The reason why we don't measure that hypoxia in monitoring programs is probably practical. I would advise on minimizing that angle in the title, intro and discussion. The bigger contribution on the manuscript is somewhere else, e.g. in the high resolution measurements and examination of oxygen drivers. If the authors decide to continue on the "hidden" hypoxia angle, I would advise on elaborating further on why does it matter that we can detect smallscale hypoxia near the sediment surface.

Response: We are happy to consider an alternative title for our paper, perhaps "Oxygen dynamics in accumulations of drifting macrophyte detritus". Our decision to focus on 'hidden hypoxia' was in fact to challenge the modus operandi of how we measure O2 in coastal waters. Even though

Jørgensen highlighted this fact in his study more than 40 years ago, we continue to measure O2 at some distance from the seabed, thus underestimating the true extent of coastal hypoxia. However, measuring O2 close to the seabed is challenging and requires some technological development. The seabed is a hotspot for biodiversity and biogeochemical cycling, so the occurrence of hypoxia should be of great interest. We will highlight this more clearly in our revision.

Update: We have revised the title and removed reference to 'hidden hypoxia' throughout the ms. The title now reads: "High metabolism and periodic hypoxia associated with drifting macrophyte detritus in the shallow subtidal Baltic Sea."

Specific comments

There should be a better distinction between the sections and experiments conducted, as the titles "O2 dynamics" (section 2.2) and "O2 fluxes" (e.g. section 2.3) are a bit confusing. To someone that is not familiar, it may seem unclear why you use oxygen sensor array in one instance and AEC in another. Please outline that better.

Response: We will provide more descriptive subheadings to better distinguish the sections.

Update: We revised the subheading of 2.2 to include square brackets around the $O_2$ (i.e. $[O_2]$), indicating that it is $O_2$ concentration we are referring to and thus distinct from 2.3: Benthic $O_2$ fluxes. We also revised subheadings 3.2 and 4.3 and include the square brackets here too. Additionally, throughout the text, all references to $O_2$ concentration now include square brackets, i.e. as $[O_2]$.

I also found it hard to know when each of the measurements were taken, and why some of the results were not included in the figures. For instance why are only 2/3 of the measurements shown in Fig. 4? It may be valid to not include some measurements, but the reader is left wondering why if no explanation is not provided. I suggest all the figures have their date of sampling included to help better guide what set of deployments the reader is looking at (e.g. oxygen array vs AEC).

Response: Thank you for catching this. We did not include the year of measurement in Fig. 4. We will include the year in the revision. In this Figure, we only show results for June and September, because these two datasets best illustrate what we want to show in the figure, i.e. the stimulated O2 consumption due to wave-driven mixing.

Update: We now exclude this Figure 4 from the revision. We do not think this figure is required. All figures now include the date, either in the figure itself or in the figure legend.

Ln. 135. Please outline better what is it that you want to measure with this technique and why.

Response: Eddy covariance integrates over a relatively large seafloor area (~30 m2) and extracts fluxes without disturbing the hydrodynamics or the light, which is particularly useful when trying to understand the mechanistic drivers of O2 dynamics. In the revision we will highlight why it is useful to measure benthic O2 fluxes using eddy covariance and what we can infer from the auto-heterotroph balance of detritus canopies.

Update: we now include a few sentences on why the eddy covariance method is appropriate for this study

L147-150: "Eddy covariance integrates over a relatively large seafloor area (typically ~30 m$^2$) (Berg et al., 2007), and extracts fluxes without disturbing the hydrodynamics or the light, which is particularly useful when trying to understand the mechanistic drivers of [O$_2$] dynamics (Berg et al., 2022)."

Ln. 163. This is a common assumption, but studies from several systems show that Rlight may be higher than Rdark. Do we know how well this assumption prevails in macroalgae systems? Including a reference may help.

Response: We do not know of studies that are specific to macroalgae canopies, but it is well documented in macrophyte canopies such as in seagrass (Juska and Berg 2022). We will mention this as well as some older studies (Fenchel and Glud, 2000) that highlight this fact.

Update: We now include a sentence stating this assumption explicitly and reference two papers

L189-190: "The latter is a common assumption but it is known that it underestimates the true metabolic activity (Fenchel and Glud, 2000; Juska and Berg, 2022)"

Ln. 260. Does that correspond to a daylight or night-time period?

Response: This is a nighttime period. We will revise the figure to include hour of day instead of deployment time.

Update: We now include the date and time in Figure 3.

Ln. 270. Personally I felt that manuscript needs to link this finding with the mat measurements better, either here in the results or discussion section.

Response: Unfortunately, we do not have measures of [O$_2$] dynamics in the canopy AND eddy covariance fluxes done simultaneously. Therefore, we can only infer what might be the case based on the two datasets, rather than link this quantitatively.

Ln. 330. How do we know that those O2 fluxes are the result of the detritus canopy and not the photosynthetic community within the sediment? A better case needs to be presented here. Consider including measurements on bare sediment area

Response: We will include measurements performed on a nearby bare sediment area. Note however that at the detritus site, the sediment was covered by a ~20 cm-thick canopy of degrading macrophyte detritus and was thus not exposed to sunlight. We will clarify this point in the revision.

Update: We now include a section in the Discussion where we compare the habitats explicitly:

L376-382: "Daily R at the detritus site was up to ~5-fold larger than that at a nearby (within 4 km) site with bare sediments and up to twice as high as a neighbouring intact canopy of *F. vesiculosus* (Attard et al., 2019b). Decaying (and respiring) fragments of *F. vesiculosus* could contribute substantially to the O$_2$ uptake rate. Laboratory incubations of *F. vesiculosus* fragments resolved respiration rates ~5 µmol O$_2$ g dw$^{-1}$ h$^{-1}$, equivalent to ~25 mmol O$_2$ m$^{-2}$ d$^{-1}$ when upscaled to *in situ* biomass observed at the detritus site (data not shown)."

Ln. 331. This is however not the main finding of the study. The Discussion would benefit from stating more upfront what the main findings of the study area in a succinct manner. E.g. you

observed hypoxia in the bottom of the mat, and you link that to mat metabolic activity combined with water flow.

Response: We will consider restructuring the Discussion although we believe this is a stylistic comment. Discussions could start with the most important finding, or they could build up to the most important finding (we currently take the latter approach).

Update: We maintained the original layout of the Discussion although we did implement several changes to better highlight the main findings of our study and their implications, as suggested by both reviewers. These are summarized below.

- In L372-382 we compare the detritus metabolism rates explicitly and infer differences about how they function from a metabolic perspective.
- In L415-422 we now highlight why we think coastal hypoxia is currently underestimated, based on the results from our present study
- In L425-430 we highlight explicitly how widespread we believe this habitat to be.

Ln. 360-370. Personally I found this paragraph a bit out of place. It seems like discussing the consequences of the hypoxia you document in the previous paragraph for faunal communities (which you measured) would flow better here.

Response: We agree that much of this information was stated in the introduction; we will largely reduce or remove this section to focus on novel outcomes.

Update: We largely reduced this section and instead focus on the areas prone to hypoxia in shallow waters, based on model predictions. The new section is reproduced below.

L410-422: "The importance of measuring [$O_2$] close to the seafloor was demonstrated more than 40 years ago by Jorgensen (1980), and since then, other researchers have investigated the distribution of dissolved constituents such as $O_2$ and nutrients in the benthic boundary layer (Holtappels et al., 2011). These studies document that solute gradients are largest near the seafloor. For practical reasons, however, coastal monitoring programs measure [$O_2$] further away from the seafloor. Models based on monitoring data suggest that hypoxia is prevalent in only small areas of the shallow subtidal zone. For instance, models for the northern Baltic Sea, which cover a total seabed area of 12435 km$^2$ of which 2211 km$^2$ is in shallow waters <5 m depth, indicate that just 16.5 km$^2$ (or 0.75% of shallow waters) are prone to hypoxia (Virtanen et al., 2019). Given that large quantities of drifting macrophytes are a common phenomenon in the shallow subtidal zone of the northern Baltic Sea (Norkko and Bonsdorff, 1996a), it is likely that coastal hypoxia is currently underestimated because large-scale models are largely based on measurements performed higher above the seafloor (0.5-1.0 m) (Virtanen et al., 2019; Conley et al., 2011)."

Ln. 380. This section is quite confusing, as it is simultaneously talking about detritus agregations, habitat structure (ln. 380), oxygen dynamics (ln. 384) and the effects of detritus mats on diversity (391). I suggest splitting it into different paragraphs. E.g. you can talk about the prevalence (seasonal, spatial) of Fucus detritus in the study area and the Baltic, the consequences of hypoxia for faunal communities, and the consequences of macroalgae-induced hypoxia for sediment communities in different paragraphs, as there is plenty to elaborate there on.

Response: Similarly, much of this information was included in the introduction and will be removed. Instead, we will talk about the prevalence of hypoxia and the generality of our findings.

Update: We removed the redundant information and added a few sentences on how widespread we expect the detritus canopies could be, as suggested by both reviewers (L425-430)

Fig. 4. It would be useful if the panels had the night-time and daytime overlayed on the deployment time axis (e.g. shaded box for night). Please consider doing that for the first panel rows of Fig. 2 and 5 as well. Also please include date of measurement

Response: OK, will implement.

Update: Based on comments from Reviewer 1 and our own evaluation we removed Figure 4. We now include date of measurement in all figures. We added grey shading to nighttime periods in Figure 5. We did not add them to Figure 2 as we didn't think this was needed.